# WASSERSTEIN DISTRIBUTIONAL NORMALIZATION : NONPARAMETRIC STOCHASTIC MODELING FOR HANDLING NOISY LABELS

## ABSTRACT

We propose a novel Wasserstein distributional normalization (WDN) algorithm to handle noisy labels for accurate classification. In this paper, we split our data into uncertain and certain samples based on small loss criteria. We investigate the geometric relationship between these two different types of samples and enhance this relation to exploit useful information, even from uncertain samples. To this end, we impose geometric constraints on the uncertain samples by normalizing them into the Wasserstein ball centered on certain samples. Experimental results demonstrate that our WDN outperforms other state-of-the-art methods on the Clothing1M and CIFAR-10/100 datasets, which have diverse noisy labels. The proposed WDN is highly compatible with existing classification methods, meaning it can be easily plugged into various methods to improve their accuracy significantly.

## 1 INTRODUCTION

The successful results of deep neural networks (DNNs) on supervised classification tasks heavily rely on accurate and high-quality label information. However, annotating large-scale datasets is extremely expensive and a time-consuming task. Because obtaining high-quality datasets is very difficult, in most conventional works, training data have been obtained alternatively using crowd-sourcing platforms Yu et al. (2018) to obtain large-scaled datasets, which leads inevitable *noisy labels* in the annotated samples.

While there are numerous methods that can deal with noisy labeled data, recent methods actively adopt the *small loss* criterion, which enables to construct classification models that are not susceptible to noise corruption. In this learning scheme, a neural network is trained using easy samples first in the early stages of training. Harder samples are then gradually selected to train mature models as training proceeds. Jiang et al. (2018) suggested collaborative learning models, in which a mentor network delivers the data-driven curriculum loss to a student network. Han et al. (2018); Yu et al. (2019) proposed dual networks to generate gradient information jointly using easy samples and employed this information to allow the networks to teach each other. Wei et al. (2020) adopted a disagreement strategy, which determines the gradient information to update based on disagreement values between dual networks. Han et al. (2020) implemented accumulated gradients to escape optimization processes from over-parameterization and to obtain more generalized results. In this paper, we tackle to solve major issues raised from the aforementioned methods based on the small-loss criterion, as follows.

In comprehensive experiments, the aforementioned methods gain empirical insight regarding network behavior under noisy labels. However, theoretical and quantitative explanation have not been closely investigated. In contrast, we give strong theoretical/empirical explanations to understand the network under noisy labels. In particular, we present an in-depth analysis of small loss criteria in a probabilistic sense. We exploit the stochastic properties of noisy labeled data and develop probabilistic descriptions of data under the small loss criteria, as follows. Let $\mathbb{P}$ be a probability measure for the pre-softmax logits of the training samples, $l$ be an objective function for classification, and $\mathbf{1}_{\{\cdot\}}$ be an indicator function. Then, our central object to deal with is a truncated measure defined as

$$X \sim \mu|\zeta = \frac{\mathbf{1}_{\{X;l(X)>\zeta\}}\mathbb{P}}{\mathbb{P}[l(X) > \zeta]}, \quad Y \sim \xi|\zeta = \frac{\mathbf{1}_{\{X;l(Y)\leq\zeta\}}\mathbb{P}}{\mathbb{P}[l(Y) \leq \zeta]}, \tag{1}$$

where $X$ and $Y$, which are sampled from $\mu|\zeta$ and $\xi|\zeta$, denote *uncertain* and *certain samples* defined in the pre-softmax feature space[1] (*i.e.*, $\mathbb{R}^d$), respectively. In equation 1, $\mu$ and $\xi$ denote the probability measures of uncertain and certain samples, respectively, and $\zeta$ is a constant. Most previous works have focused on the usage of $Y$ and the sampling strategy of $\zeta$, but poor generalization capabilities based on the abundance of uncertain samples $X$ has not been thoroughly investigated, even though these samples potentially contain important information. To understand the effect of noisy labels on the generalized bounds, we provide the concentration inequality of uncertain measure $\mu$, which renders the probabilistic relation between $\mu$ and $\xi$ and learnability of the network under noisy labels.

While most conventional methods Han et al. (2018); Wei et al. (2020); Li et al. (2019a); Yu et al. (2019) require additional dual networks to guide misinformed noisy samples, the scalability is not guaranteed due to the existence of dual architectures, which have the same number of parameters as the base network. To alleviate this problem, we build a statistical machinery, which should be fully non-parametric, simple to implement, and computationally efficient to reduce the computational complexity of conventional approaches, while maintaining the concept of small-loss criterion. Based on the empirical observation of ill-behaved certain/uncertain samples, we propose the gradient flow in the Wasserstein space, which can be induced by simulating non-parametric stochastic differential equation (SDE) with respect to the Ornstein-Ulenbeck type to control the ill-behaved dynamics. The reason for selecting these dynamics will be thoroughly discussed in the following sections.

Thus, key contributions of our work are as follows.

- We theoretically verified that there exists a *strong correlation* between model confidence and statistical distance between $X$ and $Y$. We empirically investigate that the classification accuracy worsens when the upper-bound of 2-Wasserstein distance $\mathcal{W}_2(\mu, \xi) \leq \varepsilon$ (*i.e.*, distributional distance between certain and uncertain samples) drastically increase. Due to the empirical nature of upper-bound $\varepsilon$, it can be used as an estimator to determine if a network suffers from over-parameterization.

- Based on empirical observations, we develop a *simple, non-parametric*, and *computationally efficient* stochastic model to control the observed ill-behaved sample dynamics. As a primal object, we propose the stochastic dynamics of gradient flow (*i.e.*, Ornstein-Ulenbeck process) to simulate simple/non-parametric stochastic differential equation. Thus, our method do not require any additional learning parameters.

- We provide important theoretical results. First, the controllable upper-bound $\varepsilon$ with the inverse exponential ratio is induced, which indicates that our method can efficiently control the diverging effect of Wasserstein distance. Second, the concentration inequality of transported uncertain measure is presented, which clearly renders the probabilistic relation between $\mu$ and $\xi$.

## 2 RELATED WORK

**Curriculum Learning & Small-loss Criterion.** To handle noisy labels, Han et al. (2018); Yu et al. (2019); Jiang et al. (2018); Wei et al. (2020); Lyu & Tsang (2020a); Han et al. (2020) adopted curriculum learning or sample selection frameworks. However, these methods only consider a small number of selected samples, where large portion of samples are excluded at the end of the training. This inevitably leads to poor generalization capabilities. However, this conflicts with sample selection methods because a large portion of training samples are gradually eliminated. By contrast, our method can extract useful information from unselected samples $X \sim \mu$ (*i.e.*, uncertain samples) and enhance these samples (*e.g.*, $X' \sim \mathcal{F}\mu$) for more accurate classification. Chen et al. (2019) iteratively apply cross-validation to randomly partitioned noisy labeled data to identify most samples that have correct labels. To generate such partitions, they adopt small-loss criterion for selecting samples.

**Loss Correction & Label Correction.** Patrini et al. (2017a); Hendrycks et al. (2018); Ren et al. (2018) either explicitly or implicitly transformed noisy labels into clean labels by correcting classification losses. Unlike these methods, our method transforms the holistic information from uncertain samples into certain samples, which implicitly reduces the effects of potentially noisy labels. While correction of label noisy by modifying the loss-dynamics do not perform well under extreme noise environments, Arazo et al. (2019) adopt label augmentation method called MixUp Zhang et al. (2018).

---

[1]Due to the technical difficulties, we define our central objects on pre-softmax space rather than label space, *i.e.*, the space of $\sigma(X), \sigma(Y)$, where $\sigma$ indicates softmax function. Please refer to Appendix for more details.

**Distillation.** Li et al. (2019b) updated mean teacher parameters by calculating the exponential moving average of student parameters to mitigate the impact of gradients induced by noisy labels. Lukasik et al. (2020) deeply investigated the effects of label smearing for noisy labels and linked label smoothing to loss correction in a distillation framework. Similar to these methods, our method leverages the useful properties of distillation models. We set $\nu$ as a pivot measure, which guides our normalization functional $\mathcal{F}\mu$ for uncertain measures. This is similar to self-distillation because uncertain training samples are forced to be normalized to those of past states.

**Other methods.** Lee et al. (2019) induced a robust generative classifier based on pre-trained deep models. Similar to our method, Damodaran et al. (2019) designed a constraint on the Wasserstein space and adopted an adversarial framework for classification models of noisy labeled data by implementing semantic Wasserstein distance. Pleiss et al. (2020) identify noisy labeled samples by considering AUM statistics which exploits differences in training dynamics of clean and mislabeled samples. In most recent work, Li et al. (2019a) adopts semi-supervised learning (SSL) methods to deal with noisy labels where the student network utilizes both labeled/unlabeled samples to perform semi-supervised learning guided by the other teacher network.

## 3 DISTRIBUTIONAL NORMALIZATION

Because our main target object is a probability measure (distribution), we first define an objective function in a distributional sense. Let $l$ be cross entropy and $\hat{r}$ be a corrupted label random vector for an unknown label transition matrix from a clean label $r$ which is independent of $X$, with label transition matrix $Q$. Then, a conventional objective function for classification with noisy labels can be defined as follows:

$$\min_{\mu} \mathcal{J}[\mu] = \min_{\mu} \mathbb{E}_{X \sim \mu, \hat{r}|Q}\left[l(X; \hat{r})\right]. \tag{2}$$

However, due to the significant changes in label information, the conventional objective function defined in equation 2 cannot be used for accurate classification. Instead of directly using uncertain samples $X \sim \mu$ as in previous works, we normalize $\mu$ in the form of a metric ball and present a holistic constraint. For a clear mathematical description, we first introduce the following definition.

**Definition 1.** *(Wasserstein ambiguity set) Let $\mathcal{P}_2(\mathbb{R}^d) = \{\mu : \mathbb{E}_\mu d_E^2(x_0, x) < \infty, \forall x_0 \in \mathbb{R}^d\}$ be a 2-Wasserstein space, where $d$ denotes the number of classes, $d_E$ is Euclidean distance defined on $\mathbb{R}^d$. Then, we define a Wasserstein ambiguity set (i.e., metric ball) in this space as follows:*

$$\mathbb{B}_{\mathcal{W}_2}(\nu, \varepsilon) = \left\{\mu \in \mathcal{P}_2\left(\mathbb{R}^d\right) : \mathcal{W}_2(\mu, \nu) \leq \varepsilon\right\}, \tag{3}$$

*where $\mathcal{W}_2$ denotes the 2-Wasserstein distance and $\nu$ is the pivot measure.*

Then, we propose a new objective function by imposing geometric constraints on $\mu$ as follows:

$$\min_{\mathcal{F}\mu \in \mathbb{B}_{\mathcal{W}_2}(\nu, \varepsilon), \xi} \mathcal{J}\left[\mathcal{F}\mu\right] + \mathcal{J}\left[\xi\right] = \min_{\theta} \mathbb{E}_{X \sim \mathcal{F}\mu_\theta, \hat{r}}[l(X; \hat{r})] + \mathbb{E}_{X \sim \xi_\theta, \hat{r}}[l(Y; \hat{r})], \tag{4}$$

where $\mathcal{F} : \mathcal{P}_2(\mathbb{R}^d) \to \mathcal{P}_2(\mathbb{R}^d)$ is a functional for probability measures, which assures the constraint on $\mathcal{F}\mu$ (i.e., $\mathcal{F}\mu \in \mathbb{B}_{\mathcal{W}_2}(\nu, \varepsilon)$) and our main objective. The right-hand side of equation equation 4 is equivalent vectorial form of distributional form in left-hand side. While our main objects are defined on pre-softmax, both probability measures $\mu_\theta$ and $\xi_\theta$ is parameterized by neural network with parameters $\theta$. This newly proposed objective function uses the *geometrically enhanced version* of an uncertain measure $\mathcal{F}\mu$ with a certain measure $\xi$. In equation 4, probability measure $\nu$ is defined as follows: $\nu = \arg\min \mathcal{J}[\xi_{k^\star}]$, where $\xi_k$ denotes a certain measure at the current $k$-th iteration and $k^\star \in \mathbb{I}_{k-1} = \{1, \cdots, k-1\}$. In other words, our method finds the best probability measure that represents all certain samples so far at training time, where the uncertain measures are transported to be lying in the Wasserstein ball centered on $\nu$. In equation 4, the Wasserstein constraint on $\mathcal{F}\mu$ enforces uncertain measures statistically resemble $\nu$ from a geometric perspective (i.e., $\mathcal{W}_2(\nu, \mathcal{F}\mu) \leq \varepsilon$).

Now, an important question naturally stems from the aforementioned analysis: how can we select the optimal radius $\varepsilon$? Clearly, finding an $\mathcal{F}$ that induces a small $\varepsilon \approx 0$ is suboptimal because $\mathcal{F}\mu \approx \nu$ and using objective function $\mathcal{J}[\mathcal{F}\mu \approx \nu]$ can lead to the following critical problem. As the optimization process proceeds, enhanced uncertain samples $X' \sim \mathcal{F}\mu$ contribute less and less, because it is statistically identical to $\nu$, meaning our objective in equation 4 would receive little benefits from these transported uncertain samples. By contrast, if we adopt a large radius for $\varepsilon$, enhanced uncertain samples will be statistically and geometrically unrelated to $\nu$, which causes the normalized measure $\mathcal{F}\mu$ to yield large losses and violates our objective.

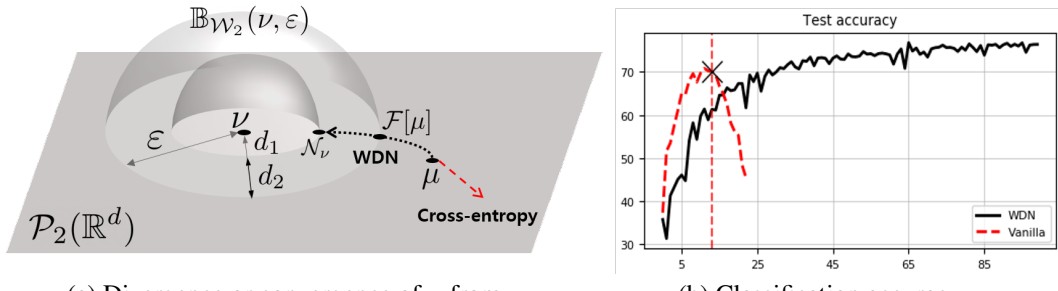

(a) Divergence or convergence of $\mu$ from $\nu$      (b) Classification accuracy

Figure 1: **Accuracy begins to drop when the uncertain measure $\mu$ begins to diverge from $\nu$.** In classification models with vanilla cross entropy losses, the uncertain measure $\mu$ can easily diverge from $\nu$ in the Wasserstein space (the red dotted line in (a)), which induces an accuracy drop (the red dotted line in (b)). By contrast, the proposed WDN can prevent such divergence by normalizing $\mu$ onto Wasserstein ambiguity set $\mathbb{B}_{\mathcal{W}_2}(\nu, \varepsilon)$ (the black dotted line in (a)) and can consistently enhance accuracy as iteration proceeds (the black line in (b)). Please note that $d_1$ and $d_2$ denote the first and second terms in equation 5, respectively, and $\varepsilon = d_1 + d_2$.

To overcome two problems above and select the radius, we make a detour, *i.e.*, a Gaussian measure, for cutting the path between $\nu$ and $\mathcal{F}\mu$ (*i.e.*, $\nu \to \mathcal{N}(\mathbf{m}_\nu, \Sigma_\nu) \to \mathcal{F}\mu$) rather than directly calculating the geodesic between $\nu$ and $\mathcal{F}\mu$ (*i.e.*, $\nu \to \mathcal{F}\mu$). Specifically, we decompose the original constraint in equation 4 into two terms using the triangle inequality of the Wasserstein distance:

$$\mathcal{W}_2\left(\nu, \mathcal{F}\mu\right) \leq \varepsilon = \underbrace{\mathcal{W}_2\left(\nu, \mathcal{N}(\mathbf{m}_\nu, \Sigma_\nu)\right)}_{d_1:\text{ Intrinsic statistics}} + \underbrace{\mathcal{W}_2\left(\mathcal{N}(\mathbf{m}_\nu, \Sigma_\nu), \mathcal{F}\mu\right)}_{d_2:\text{ Wasserstein Normalization}}. \tag{5}$$

The first *intrinsic statistics* term sets a detour point as a Gaussian measure, for which the mean and covariance are the same as those for $\nu$ (*i.e.*, $\mathbf{m}_\nu = \mathbb{E}_{Y \sim \nu}[Y]$ and $\Sigma_\nu = \mathbf{Cov}_{Y \sim \nu}[Y]$). The Wasserstein upper bound of this term is only dependent on the statistical structure of $\nu$ because $(\mathbf{m}_\nu, \Sigma_\nu)$ is dependent on $\nu$. Thus, this term induces a **data-dependent, non-zero constant** upper bound whenever $\nu \neq \mathcal{N}$ and can prevent the upper-bound from collapsing to $\varepsilon \to 0$, regardless of $\mathcal{F}$. This gives huge advantage when dealing with $\varepsilon$ because the first term can be considered a fixed constant during the training. The second *normalization* term represents our central objective. $\mathcal{F}$ facilitates geometric manipulation in the Wasserstein space and prevent uncertain measure $\mu$ from diverging, where $\mu$ is normalized onto the Wasserstein ambiguity $\mathbb{B}_{\mathcal{W}_2}(\nu, \varepsilon)$ in Fig1. The theoretical/numerical advantages of setting detour measure as Gaussian is well-explained following section.

### 3.1 WASSERSTEIN NORMALIZATION

In the previous section, we present a novel objective function that imposes a geometric constraint on $\mu$ such that the transformed measure $\mathcal{F}\mu$ lies in $\mathbb{B}_{\mathcal{W}_2}(\nu, \varepsilon)$ for $\nu$. Now, we specify $\mathcal{F}$ and relate it to the Gaussian measure (generally Gibbs measure). For simplicity, we denote $\mathcal{N}_\nu = \mathcal{N}(\mathbf{m}_\nu, \Sigma_\nu)$.

**Proposition 1.** *$\mathcal{F} : \mathbb{R}^+ \times \mathcal{P}_2 \to \mathcal{P}_2$ is a functional on the probability measure such that $\mathcal{F}[t, \mu] = \mu_t$, where $d\mu_t = p_t d\mathcal{N}_\nu, d\mathcal{N}_\nu = dq_t dx$, and $\mu_t$ is a solution to the following continuity equations:*

$$\partial_t \mu_t = \nabla \cdot (\mu_t v_t), \tag{6}$$

*which is read as $\partial_t p(t, x) = \nabla \cdot (p(t, x) \nabla \log q(t, x))$ in a distributional sense. Then, a uniquely defined functional $\mathcal{F}_t[\cdot] = \mathcal{F}[t, \cdot]$ normalizes $\mu$ onto $\mathbb{B}_{\mathcal{W}_2}(\mathcal{N}_\nu, e^{-t}K_2(\mu))$, where $K_2(\mu) > 0$ is a constant that depends on $\mu$.*

It is well known that the solution to equation 6 induces a geodesic in the 2-Wasserstein space (Villani (2008)), which is the shortest path from $\mu = \mu_{t=0}$ to $\mathcal{N}_\nu$. The functional $\mathcal{F}_t$ generates a path for $\mu_t$, in which the distance is exponentially decayed according to the auxiliary variable $t$ and constant $K_2$, meaning $\mathcal{W}_2(\mathcal{N}_\nu, \mathcal{F}_t\mu) \leq K_2 e^{-t}$. This theoretical results indicates that the Wasserstein distance of second term in equation 5 can be **reduced/controlled with exponential ratio**. Thus, by setting a different $t$, our method can efficiently control the diverging distance in equation 5. Unfortunately, it is typically intractable to compute the partial differential equation (PDE) in equation 6.

---

**Algorithm 1** Wasserstein Distributional Normalization

---

**Require:** $\alpha \in [0, 0.2], \varrho \in [0.1, 0.65], T = 64, \Delta_t = 10^{-4}, \tau = 0.001,$
   **for** $k = 1$ to $K$ (*i.e.*, the total number of training iterations) **do**
      **1)** Select uncertain $(1 - \rho)N$ and certain $\rho N$ samples from the mini-batch $N$.
      $\{Y_k^n\}_{\{n \leq \rho N\}} \sim \xi_k, \{X_k^n\}_{\{n \leq (1-\rho)N\}} \sim \mu_k$
      **2)** Update the most certain measure $\nu$.
      **if** $\mathcal{J}[\xi_k] < \mathcal{J}[\nu]$ **then**
         $\nu \leftarrow \xi_k, \mathbf{m}_\nu \leftarrow \mathbb{E}[Y_k],$ and $\Sigma_\nu \leftarrow \mathbf{Cov}[Y_k]$
      **end if**
      **3)** Update the moving geodesic average $\mathcal{N}(\mathbf{m}^\alpha, \Sigma^\alpha)$.
      Solve the Ricatti equation $\mathcal{T}\Sigma_\nu\mathcal{T} = \Sigma_{\xi_k}$.
      $\Sigma^\alpha = ((1 - \alpha)\mathbf{I}_d + \alpha\mathcal{T}) \Sigma_\nu ((1 - \alpha)\mathbf{I}_d + \alpha\mathcal{T})$ and $\mathbf{m}^\alpha = (1 - \alpha)\mathbf{m}_\nu + \alpha\mathbf{m}_{\xi_k}$
      **4)** Simulate the discrete SDE for $T$ steps.
      **for** $t = 0$ to $T - 1$ **do**
         $X_{k,t+1}^n = -\nabla\phi(X_{k,t}^n; \mathbf{m}^\alpha)\Delta_t + \sqrt{2\tau^{-1}}\Sigma_\nu^\alpha dW_t^n \ s.t. \ \{X_{k,t=0}^n\} \sim \mu_k, \{X_{k,t=T}^n\} \sim \mathcal{F}_T\mu_k$
      **end for**
      **5)** Update the network with the objective function.
      $\mathcal{J}[\mathcal{F}\mu_k] + \mathcal{J}[\xi_k] = \mathbb{E}_{\mathcal{F}_T\mu_k}[l(X_{k,T}; \hat{r})] + \mathbb{E}_{\xi_k}[l(Y_k; \hat{r})]$
   **end for**

---

To solve this problem, we adopt particle-based stochastic dynamics, which enables tractable computation. There exists a unique iterative form corresponding PDE in equation 6 which is called as *multi-dimensional Ornstein-Ulenbeck process*, which can be approximated using particle-based dynamics. In particular, we draw $N(1 - \varrho)$ uncertain samples from a single batch of $N$ samples using equation 1 for hyper-parameter $0 \leq \varrho \leq 1$. We then simulate a discrete stochastic differential equation (SDE) for each particle using the Euler-Maruyama scheme as follows:

$$X_{t+1}^n = X_t^n - \nabla\phi(X_t^n; \mathbf{m}_\nu)\Delta_t + \sqrt{2\tau^{-1}\Delta_t\Sigma}Z_{\mathbf{I}}^n, \tag{7}$$

where $\phi(X_t; \mathbf{m}_\nu) = \frac{\tau}{2}d_E^2(X_t, \mathbf{m}_\nu), n \in \{1 \cdots, N(1 - \varrho)\}$, $d_E$ is a Euclidean distance, and $N$ is a single mini-batch size. We selected OU process as our stochastic dynamic due to the following reasons: First, we want to build computationally efficient, and non-parametric method to estimate/minimize the second term of equation 5. The SDE in equation 7 corresponding OU process have simple form with fixed drift and diffusion terms which is invariant over times which makes us to induce the non-parametric representations of simulation of SDE. While the simulation of equation 7 is just non-parametric for-loops in implementation algorithm, our method is **computationally very efficient** compared to other baseline methods such as Han et al. (2018). Second, when estimating empirical upper-bound of Wasserstein distance, OU process allows us to use explicit form called *Meheler's formula* which can be efficiently estimated (Please refer to Appendix for more details). The overall procedure for our method is summarized in Algorithm 1.

## 3.2 WASSERSTEIN MOVING GEODESIC AVERAGE

In our experiments, we observe that the best measure $\nu$ is not updated for a few epochs after the training begins. This is problematic because $\nu$ diverges significantly from the current certain measure $\xi_k$, which is equivalent to the normalized measure $\mathcal{F}\mu_k$ diverging from $\xi_k$, meaning $X_T$ and $Y$ become increasingly statistically inconsistent. To alleviate this statistical distortion, we modify detour measure from $\mathcal{N}_\nu$ to other Gaussian measure, which allows us to capture the statistics of both $\xi_k$ and $\nu$. Inspired by the moving average of Gaussian parameters in batch normalization Ioffe & Szegedy (2015), we propose the *Wasserstein moving geodesic average*. Specifically, we replace Gaussian parameters $\{\mathbf{m}_\nu, \Sigma_\nu\}$ with $\{\mathbf{m}^\alpha, \Sigma^\alpha\}$ such that $\mathbf{m}^\alpha = (1 - \alpha)\mathbf{m}_\nu + \alpha\mathbf{m}_{\xi_k}$ and $\Sigma^\alpha = ((1 - \alpha)\mathbf{I}_d + \alpha\mathcal{T})\Sigma_\nu((1 - \alpha)\mathbf{I}_d + \alpha\mathcal{T})$, where $\mathcal{T}$ is a solution to the Riccati equation $\mathcal{T}\Sigma_\nu\mathcal{T} = \Sigma_{\xi_k}$. Therefore our final detour Gaussian measure is set to $\mathcal{N}_\nu^\alpha := \mathcal{N}(\mathbf{m}(\alpha), \Sigma(\alpha)), 0 \leq \alpha \leq 1^2$.

## 4 THEORETICAL ANALYSIS

In equation 5, we select the detour point as a Gaussian measure because this measure can provide a statistical structure, which is similar to that of the optimal $\nu$. In addition to this heuristic motivation, setting a detour point as a Gaussian measure (Gibbs measure) also provides theoretical advantages, *e.g.*, the theoretical upper bound of the Wasserstein constraint terms. In this section, we investigate the explicit upper bounds of two terms in equation 5, which are naturally induced by the SDE.

---

[2]Please refer to Appendix C.4 for more details.

**Proposition 2.** *A scalar $0 < \beta < \infty$ exists and depends on $\nu$, resulting in the following inequality:*

$$\mathcal{W}_2(\nu, \mathcal{F}_t\mu) \leq \varepsilon = K_1(\nu) \vee \left[e^{-t}K_2(\mu) + K_2(\nu)\right], \tag{8}$$

*where $\lambda_{\max}(\Sigma_\nu)$ denotes the maximum eigenvalue of the covariance matrix $\Sigma_\nu$ and for some constant $0 < K_1 < \infty$, we have $K_1(\nu) = \sqrt{d\beta\lambda_{max}(\Sigma_\nu)} + \|\mathbb{E}_\nu Y\|_2$ which is only dependent on $\nu$.*

Intuitively, $K_2(\mu)$ can be interpreted as an indicator that tells us how the uncertain measure $\mu$ is diffused, whereas the designed term $e^{-t}K_2(\mu)$ controls the upper bound of the Wasserstein distance using a variable $t$. The other term $K_2(\nu)$ does not vanish even with a very large $t$, which assures a non-collapsing upper-bound $\varepsilon$.

**Proposition 3.** *(**Concentration inequality for the normalized uncertain measure**). Assume that there are some constants $T \in [\frac{1}{\eta}, \infty), \eta \geq 0$ such that the following inequality holds:*

$$\mathbb{E}_{\mathcal{F}_T\mu}[f^2] - [\mathbb{E}_{\mathcal{F}_T\mu}[f]]^2 \leq (1+\eta)\mathbb{E}_{\mathcal{F}_T\mu}[\mathbf{A}\nabla f^T \nabla f], \quad f \in C_0^\infty(\mathbb{R}^d), \tag{9}$$

*for $\mathbf{A} \in \mathbf{Sym}_d^+$ and $D(\mathbf{A}, \Sigma_\nu) \leq a\eta$ for some $a > 0$ with any metric $D$ defined on $\mathbf{Sym}_d^+$. In this case, there is a $\delta$ such that the following probability inequality for an uncertain measure is induced:*

$$\mathcal{F}_T\mu\left(|\sigma - \mathbb{E}_\nu[\sigma]| \geq \delta\right) \leq 6e^{-\frac{\sqrt{2}\delta^{\frac{3}{2}}}{K_2(\mu)}}, \tag{10}$$

*where $\sigma$ denotes a soft-max function.*

In equation 10, we show that the label information induced by the normalized uncertain measure is close to that of most certain measure $\mathbb{E}_\nu[\sigma]$, where the upper bound is exponentially relative to the initial diffuseness of $\mu$ (*i.e.*, $K_2(\mu)$). Because the upper bound of the probability inequality does not collapse to zero and $\mathcal{F}_T\mu$ is concentrated around the most certain labels (*i.e.*, $\mathbb{E}_\nu[\sigma]$), the uncertain sample $X_T \sim F_T\mu$ helps our method avoid over-parameterization.

## 4.1 EMPIRICAL UNDERSTANDINGS

We investigate the theoretical upper bound of the Wasserstein ambiguity (*i.e.*, radius of the Wasserstein ball) for $\mathcal{F}\mu$ and its corresponding probability inequality. To provide more in-depth insights into the proposed method, we approximate the upper bound and demonstrate that our Wasserstein normalization actually makes neural networks more robust to label noise.

As we verified previously, according to Proposition 2, the following inequality holds:

$$\mathcal{W}_2(\mathcal{F}_t\mu, \nu) \leq \varepsilon = K_1(\nu) \vee (K_2(\nu) + K_2(\mathcal{F}_t\mu)). \tag{11}$$

Because the first term $K_1(\nu)$ is constant, dependent on $\nu$, and generally small compared to the second term with $t \leq T$, we only examine the behavior of the second term $K_2(\nu) + K_2(\mathcal{F}_t\mu)$, which can be efficiently approximated using a simple form. Because our detour measure is Gaussian, we have the following inequality for any $h \in C_0^\infty(\mathbb{R}^d)^3$:

$$\hat{K}_2(\mu) = \lim_{s \to 0} \frac{1}{s}\mathbb{E}_{X,Z \sim \mathcal{N}_{\mathbf{I}}}\left[h\left(e^{-s}X + \sqrt{1 - e^{-2s}}(\Sigma_\nu^{\frac{1}{2}}Z + \mathbf{m}_\nu)\right) - h(X)\right] \leq K_2(\mu), \tag{12}$$

where this equality holds if $h$ is selected to induce a supremum over the set $C_0^\infty$. For approximation, we simply consider $h(X) = \|X\|_2$ as a test function. In this case, the following inequality naturally holds: $\hat{\varepsilon} = \hat{K}_2(\nu) + \hat{K}_2(\mathcal{F}\mu) \leq K_2(\nu) + K_2(\mathcal{F}\mu) \leq K_1(\nu) \vee (K_2(\nu) + K_2(\mathcal{F}\mu)) = \varepsilon$. Thus, $\hat{\varepsilon}$ can be considered as an approximation of the theoretical upper bound $\varepsilon$ suggested in Proposition 2. Subsequently, we investigate the effects of Wasserstein normalization based on $\hat{K}_2(\mu)$ in equation 12.

**(1) The proposed WDN ensures that the Wasserstein ambiguity is bounded.** We examine the relation between $\hat{\varepsilon}$ and test accuracy in an experiment using the CIFAR-10 dataset with symmetric noise at a ratio of $0.5$. Fig.2 presents the landscape for the $\log_{10}$-scaled cumulative average of $\hat{\varepsilon}$ and test accuracy over epochs. The red dotted lines represent the landscape of the vanilla network with cross-entropy loss, where $\hat{\varepsilon}_k = \hat{K}_2(\nu_k) + \hat{K}_2(\mathcal{F}_{t=0}\mu_k)$ and $k$ is the epoch index. In this case, the time constant $t$ is set to zero, because Wasserstein normalization is not employed for the vanilla network. The black lines indicate the landscape of the proposed method, where $\hat{\varepsilon}_k = \hat{K}_2(\nu_k) + \hat{K}_2(\mathcal{F}_{t=T}\mu_k)$

---
[3]Please refer to Appendix C.2 for additional details.

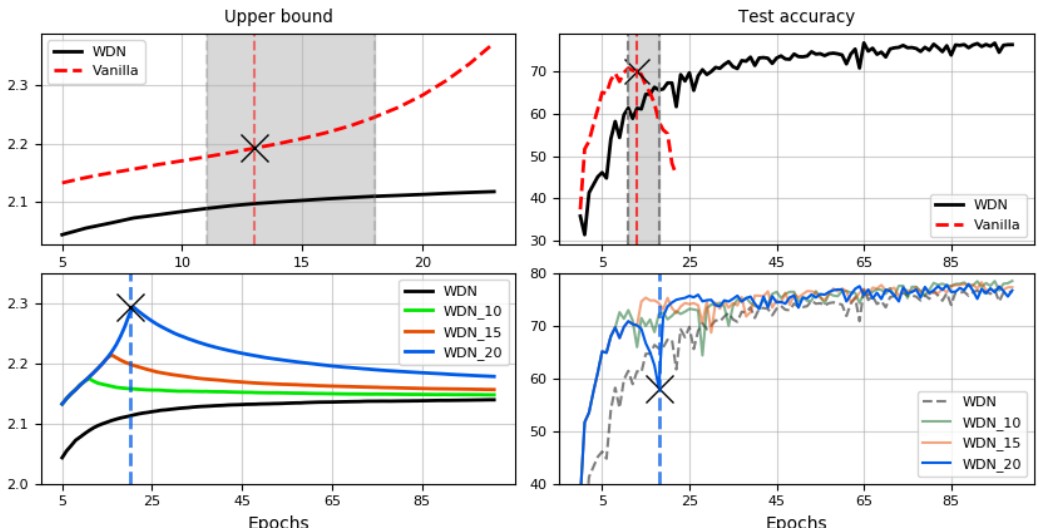

Figure 2: **Relation between the approximated upper bound $\hat{\varepsilon}$ and test accuracy**.

in this case. It is noteworthy that the test accuracy of the vanilla network begins to decrease after 13-epochs (red-dotted vertical lines in the top-right plot), whereas the *Wasserstein ambiguity (i.e., upper bound of the Wasserstein distance) increases quadratically* in the top-left plot. These experimental results verify that the distance between uncertain and most certain measure (*i.e.*, $\nu$) becomes large in the 2-Wasserstein space without any constraints in vanilla networks. They also indicate a definite relationship between Wasserstein ambiguity and test accuracy. In the proposed WDN, Wasserstein ambiguity can be efficiently bounded (*i.e.*, $\limsup_k \hat{\varepsilon}_k \approx 2.15$) as the test accuracy continues to increase, even after 13-epochs. For detailed analysis, we compute the deviation of an empirical upper bound as follows: $\hat{\Delta}_k = \hat{\varepsilon}_k - \hat{\varepsilon}_{k-1}$. In the gray regions, the deviation for the vanilla network is grater than $2.5 \times 10^{-2}$, *i.e.*, $\Delta_k > 2.5 \times 10^{-2}$. Then, its test accuracy begins to drop, as shown in Fig.2. In contrast to the vanilla network, the maximum deviation of the proposed WDN is bounded above by a very small value ($\sup_k \hat{\Delta}_k \leq 8 \times 10^{-3}$).

**(2) The proposed WDN helps networks to escape from over-parameterization.** To analyze the behavior of deep neural networks under over-parameterization with and without the proposed WDN, we design several variants of the WDN, which begin at delayed epochs. The green, orange, and blue curves in the second row of Fig.2 represent the landscapes, when our WDN is applied after $k_d \in \{10, 15, 20\}$ epochs, respectively. In this experiment, the upper bound $\hat{\varepsilon}_k$ is defined as

$$\hat{\varepsilon}_k = \begin{cases} \hat{K}_2(\nu_k) + \hat{K}_2(\mathcal{F}_{t=0}\mu_k), & \text{if } k < k_d, \\ \hat{K}_2(\nu_k) + \hat{K}_2(\mathcal{F}_{t=T}\mu_k), & \text{else } k \geq k_d. \end{cases} \qquad (13)$$

Consider $k_d = 20$, which is represented by the blue dotted vertical lines. Before our WDN is applied (*i.e.*, $k < k_d$), the network suffers from over-parameterization, which induces a significant performance drop, as indicated by the blue curve in the bottom-right plot. However, the network rapidly recovers to normal accuracy following Wasserstein normalization (*i.e.*, $k \geq k_d$). Please note that similar behavior can be observed in the green and orange curves. In particular, the orange curve produces less fluctuations than the blue curve in terms of test accuracy. This indicates that the proposed WDN can help a network escape from over-parameterization by imposing geometric constraints on the Wasserstein space with proposed method.

**(3) The proposed WDN can derive data-dependent bounds according to different noise levels.** Another interesting point in Fig.2 is that all curves, excluding the red curve, converge to specific numbers $2.15 = \underline{\varepsilon} := \liminf_k \hat{\varepsilon}_k \leq \limsup_k \hat{\varepsilon}_k := \bar{\varepsilon} = 2.2$. The upper bound $\bar{\varepsilon}$ is neither overly enlarged nor collapsed to zero, while the lower bound $\underline{\varepsilon}$ is fixed for all curves. We argue that this behavior stems from the geometric characteristics of the proposed method, where the first term in equation 5, namely $\mathcal{W}_2(\nu, \mathcal{N}_\nu) \propto \hat{K}_2(\nu)$, is a non-zero data-dependent term that is minimized by the proposed geometric constraint. Therefore, we can derive the following relationship:

$$[\mathcal{W}_2(\nu, \mathcal{F}\mu) \leq \mathcal{W}_2(\nu, \mathcal{N}_\nu) + \mathcal{W}_2(\mathcal{N}_\nu, \mathcal{F}\mu)]\Downarrow \;\propto\; [\hat{K}_2(\nu) + \hat{K}_2(\mathcal{F}\mu) = \hat{\varepsilon}]\Downarrow. \qquad (14)$$

This empirical observation verifies that a detour point, which is set as a Gaussian measure, *can induce the data-dependent bound* $(\underline{\varepsilon}, \bar{\varepsilon})$, where our data-dependent bound can vary according to different

Table 1: **Average test accuracy (%) on the CIFAR-10/100 dataset over the last 10 epochs with various noise corruptions.** The symbol $^\star$ indicates scores provided by the corresponding authors. $\text{WDN}_{cot}$ denotes our WDN combined with a co-teaching network. The best results are boldfaced.

| Methods | Symmetric 20% | Symmetric 50% | Asymmetric 45% |
|---|---|---|---|
| Vanilla | $71.91 \pm .43/40.44 \pm .36$ | $49.54 \pm .41/21.34 \pm .27$ | $49.06 \pm 1.02/31.85 \pm .85$ |
| MentorNet$^\star$ | $80.76 \pm .36/52.13 \pm .40$ | $71.10 \pm .48/39.00 \pm 1.00$ | $58.14 \pm .38/31.60 \pm .51$ |
| GCE | $84.68 \pm .05/51.86 \pm .09$ | $61.80 \pm .11/37.60 \pm .08$ | $61.09 \pm .18/33.13 \pm .14$ |
| RoG$^\star$ | $84.32$ / $58.16$ | $76.67$ / $45.42$ | $71.26$ / $43.18$ |
| JoCoR | $85.73 \pm .19 /53.01 \pm .04$ | $79.41 \pm .25 /43.49 \pm .46$ | $64.21 \pm .12/26.51 \pm .32$ |
| NPCL$^\star$ | $84.30 \pm .07/55.30 \pm .09$ | $77.66 \pm .09/42.56 \pm .06$ | − |
| SIGUA$^\star$ | $\leq 84$ / − | $\leq 78$ / − | $\leq 65$ / − |
| DivideMix | − | $81.13 \pm .18$ / $\mathbf{49.41 \pm .25}$ | $68.93 \pm .33$ / $34.24 \pm .63$ |
| WDN | $\mathbf{87.40 \pm .23} /\mathbf{59.18 \pm .29}$ | $82.89 \pm .13/48.45 \pm .27$ | $\mathbf{76.12 \pm .29}/38.23 \pm .31$ |
| Co-teaching | $78.23 \pm .27/53.89 \pm .09$ | $72.81 \pm .20/34.96 \pm .50$ | $70.46 \pm .58/34.55 \pm .12$ |
| Co-teaching$^+$ | $80.64 \pm .15/56.15 \pm .09$ | $58.43 \pm .30/37.88 \pm .06$ | $70.78 \pm .11/32.88 \pm .25$ |
| WDN$_{cot}$ | $\mathbf{87.12 \pm .16/57.27 \pm .33}$ | $\mathbf{76.06 \pm .28/42.38 \pm .28}$ | $\mathbf{74.11 \pm .35/44.41 \pm .37}$ |

Table 2: **Test accuracy on the CIFAR-10 dataset with open-set noisy labels from CIFAR-100**.

| Methods | Vanilla | GCE | Co-teaching | Co-teaching$^+$ | JoCoR | WDN |
|---|---|---|---|---|---|---|
| Accuracy | 38.12 | 46.57 | 35.77 | 42.57 | 47.73 | **51.28** |

noise levels and efficiently leverage data-dependent statistics. Fig.2 indicates that classification models with more stable data-dependent bounds also induce more stable convergence in test accuracy.

# 5 EXPERIMENTS

## 5.1 EXPERIMENTS ON THE CIFAR-10/100 DATASET

We used settings similar to those proposed by Laine & Aila (2016); Han et al. (2018) for our experiments on the CIFAR10/100 dataset. We used a 9-layered CNN as a baseline architecture with a batch size of 128. We used the Adam optimizer with $(\beta_1, \beta_2) = (0.9, 0.99)$, where the learning rate linearly decreased from $10^{-3}$ to $10^{-5}$.

**Synthetic Noise.** We injected label noise into clean datasets using a noise transition matrix $Q_{i,j} = \mathbf{Pr}(\hat{r} = j | r = i)$, where a noisy label $\hat{r}$ is obtained from a true clean label $r$. We defined $Q_{i,j}$ by following the approach discussed by Han et al. (2018). For symmetric noise, we used the polynomial, $\varrho = -1.11r^2 + 1.78r + 0.04$ for $0.2 \leq r \leq 0.65$, where $r$ is the noise ratio. For the asymmetric noise, we set $\varrho$ to 0.35. To select the enhanced detour measure, we set $\alpha$ to 0.2 for the Wasserstein moving geodesic average in all experiments. We trained our classification model over 500 epochs because the test accuracy of our method continued increasing, whereas those of the other methods did not. We compared our method with other state-of-the-art methods, including [MentorNet, Jiang et al. (2018)], [Co-teaching, Han et al. (2018)], [Co-teaching+, Yu et al. (2019)], [GCE, Zhang & Sabuncu (2018)], [RoG, Lee et al. (2019)], [JoCoR, Wei et al. (2020)], [NPCL, Lyu & Tsang (2020b)], [SIGUA, Han et al. (2020)], and [DivideMix, Li et al. (2019a)]. As shown in Table 1, the proposed WDN significantly outperformed other baseline methods. Please note that our WDN utilizes a simple Gaussian measure as a target pivot measure. Thus, there are potential risks when handling highly concentrated and non-smooth types of noise (*e.g.*, asymmetric noise). Nevertheless, the proposed WDN still produced accurate results, even with asymmetric noise. In this case, a variant of our WDN (*i.e.*, WDN$_{cot}$) exhibited the best performance.

**Open-set Noise.** In this experiment, we considered the open-set noisy scenario suggested by Wang et al. (2018), where a large number of training images were sampled from other CIFAR-100 dataset; however, these images were still labeled according to the classes in the CIFAR-10 dataset. We used a 9-layered CNN, which also used in our previous experiment. For hyper-parameters, we set $\varrho$ and $\alpha$ to 0.5 and 0.2, respectively. As shown in Table 2, our method achieved state-of-the-art accuracy.

**Collaboration with Other Methods.** Because our core methodology is based on small loss criteria, our method can collaborate with co-teaching methods. In Han et al. (2018), only certain samples $(Y \sim \xi)$ were used for updating colleague networks, where the number of uncertain samples gradually decreased until it reached a predetermined portion. To enhance potentially bad statistics for co-teaching, we taught dual networks by considering a set of samples $(Y, X_T)$, where $X_T \sim \mathcal{F}_T \mu$ are uncertain samples enhanced using equation 7.

Figure 3: **Test accuracy for the proposed collaboration model with co-teaching.**

Table 1 shows the test accuracy results for the proposed collaboration model with a co-teaching network (WDN$_{cot}$). This collaboration model achieved the most accurate performance for the CIFAR-100 dataset with asymmetric noise, which verifies that our WDN can be integrated into existing methods to improve their performance significantly, particularly when the density of pre-logits is highly-concentrated. Fig.3 reveals that co-teaching quickly falls into over-parameterization and induces drastic drop in accuracy after the 15th-epoch. WDN$_{cot}$ also exhibits a slight accuracy drop. However, it surpassed the baseline co-teaching method by a large margin ($+7\%$) during training. This demonstrates that our enhanced samples $X_T$ can alleviate the over-parameterization issues faced by conventional co-teaching models, which helps improve their accuracy significantly.

### 5.2 EXPERIMENTS ON A REAL-WORLD DATASET

To evaluate our method on real-world datasets, we employed the Clothing1M dataset presented by Xiao et al. (2015), which consists of 1M noisy, labeled, and large-scale cloth images with 14 classes collected from shopping websites. It contains 50K, 10K, and 14K clean images for training, testing, and validation, respectively. We only used a *noisy* set for training; for testing, we used a *clean* set. We set $\alpha = 0.2$ and $\varrho = 0.1$. For fair comparison, we followed the settings suggested in previous works. We used a pre-trained ResNet50 for a baseline architecture with a batch size of $48$. For the pre-processing steps, we applied a random center crop, random flipping, and normalization to $224 \times 224$ pixels. We adopted the Adam optimizer with a learning rate starting at $10^{-5}$ that linearly decayed to $5 \times 10^{-6}$ at 24K iterations. Regarding the baseline methods, we compared the proposed method to [GCE, Zhang & Sabuncu (2018)], [D2L, Ma et al. (2018)], [FW, Patrini et al. (2017b)], [WAR, Damodaran et al. (2019)], [SL, Wang et al. (2019)], [JOFL, Tanaka et al. (2018)], [DMI, Xu et al. (2019)], [PENCIL, Yi & Wu (2019)], and [MLNT, Li et al. (2019b)]. Table 3 reveals that our method achieved competitive performance as comparison with other baseline methods.

Table 3: **Test accuracy (mean, %) on the Clothing 1M dataset.**

| Methods | GCE | D2L | FW | JoCoR | WAR | SL | JOFL | DMI | MLNT | PENCIL | WDN | DivideMix |
|---|---|---|---|---|---|---|---|---|---|---|---|---|
| Accuracy | 69.0 | 69.47 | 69.84 | 70.30 | 70.66 | 71.02 | 72.23 | 72.46 | 73.47 | 73.49 | 74.75 | 74.76 |

### 5.3 COMPUTATIONAL COST

Because Co-teaching, JoCoR, and DivideMix use additional networks, the number of network parameters is twice ($8.86M$) as many as that of the Vanilla network ($4.43M$). In Table 4, we compare the average training time for first 5-epochs over various baseline methods under symmetric noise on the CIFAR-10 dataset. While non-parametric methods such as GCE and WDN require less than $12\%$ additional time, other methods that require additional networks spent more time than non-parametric methods. The averaging time can vary according to different experimental environments. In table 4, we measure the time using publicly available code provided by authors.

Table 4: **Average training time for the 5-epochs (sec) on the CIFAR-10 dataset.**

| Methods | Vanilla | GCE | WDN | Co-teaching | JoCoR | DivideMix |
|---|---|---|---|---|---|---|
| Time | $11.43 \pm .05$ | $11.53 \pm .06$ | $12.72 \pm .08$ | $15.88 \pm .11$ | $17.88 \pm .11$ | $34.41 \pm .53$ |
| $\Delta$ | - | $+9\%$ | $+11.3\%$ | $+38.9\%$ | $+56.3\%$ | $+201\%$ |

## 6 CONCLUSION

We proposed a novel method called WDN for accurate classification of noisy labels. The proposed method normalizes uncertain measures to data-dependent Gaussian measures by imposing geometric constraints in the 2-Wasserstein space. We simulated discrete SDE using the Euler-Maruyama scheme, which makes our method fast, computationally efficient, and non-parametric. In theoretical analysis, we derived the explicit upper-bound of the proposed Wasserstein normalization and experimentally demonstrated a strong relationship between this upper-bound and the over-parameterization. We conducted experiments both on the CIFAR-10/100 and Clothing1M datasets. The results demonstrated that the proposed WDN significantly outperforms other state-of-the-art methods.

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

## A    OPEN-SOURCE DATASET

**Transition matrix for CIFAR-10/100.** For the experiment summarized in Table 1, we implemented open-source code to generate the noise transition matrix discussed by Han et al. (2018), as well as the 9-layered CNN architecture (`https://github.com/bhanML/Co-teaching`).

**Open-set noise.**    For the experiment summarized in Table 2, we used the same dataset for open-set noisy labels presented by Lee et al. (2019) (`https://github.com/pokaxpoka/RoGNoisyLabel`).

**Clothing1M.** For the experiment summarized in Table 3, we used the open-source dataset presented by Xiao et al. (2015) (`https://github.com/Cysu/noisy_label`).

## B    COMPARISONS TO RELATED WORKS

| Methodology | Parametric | Class-dependency | Distillation | Sample-weight | Sample-selection |
|---|---|---|---|---|---|
| DivideMix | ✓ | ✗ | ✗ | ✗ | ✓ |
| Co-teaching | ✓ | ✗ | ✓ | ✗ | ✓ |
| JoCoR | ✓ | ✗ | ✓ | ✗ | ✓ |
| MLNT | ✓ | ✓ | ✓ | ✗ | ✗ |
| Ren et al. (2018) | ✗ | ✗ | ✗ | ✓ | ✗ |
| NPCL | ✗ | ✗ | ✗ | ✓ | ✗ |
| GCE | ✗ | ✗ | ✗ | ✓ | ✗ |
| WDN | ✗ | ✗ | ✗ | ✗ | ✗ |

Table B indicates that no previous methodologies can conceptually include our method.

Because the solution to the Fokker-plank equation can be explicitly calculated without any additional parameters, our method is fully non-parametric (in terms of additional parameters beyond those required by the original neural network). By contrast, co-teaching is parametric because it requires a clone network with additional parameters that are copies of those in the original network. Similarly, MLNT requires an additional teacher network for training, which also contains a number of parameters.

Many method based on small loss criteria select certain samples, whereas our method uses the combination of $\rho N$ certain and $(1 - \rho)N$ normalized uncertain samples. Therefore, our method can fully leverage the batches of training datasets, where $(1 - \rho)N + \rho N = N$. Additionally, our method does not assume any class-dependent prior knowledge. Rather than considering class-wise prior knowledge, our method uses holistic information from both certain and uncertain samples (*i.e.*, $Y$ and $X_T$ ) in the logit space. Other meta-class-based model, such as MLNT, assume class-wise meta prior knowledge from a teacher network.

In Arazo et al. (2019), they assumed the beta-mixture model as a label distribution on label space. But due to the non-deterministic type of noisy label distribution, it sometimes fails to train with extremely non-uniform type of noise. For example, Arazo et al. (2019) reported failure case with Clothing1M dataset. It seems that fundamental assumption on noise model of *mixup* will be improved in future work. Similar to this method, our work have trouble when dealing with synthetic asymmetric noise with high ratio where relatively large performance drop is observed in Table 1 (despite our method produces second best performance in the table).

Most recent work Li et al. (2019a), they also adopt Co-train by implementing additional dual network, but much sophisticated methodology called Co-divide/guessing based on SSL. We predict that the Wasserstein distance between labeled and unlabeled probability measures is well-controlled in their method. We think that applying the OT/Markov theory (as in our paper) to their method will broaden the understanding of LNL problem.

In contrast to sample weight methods such as GCE and NPCL, which require prior knowledge regarding the cardinality of the training samples to be weighted, our method is free from such assumptions because our Wasserstein normalization is applied in a batch-wise manner.

## C    TECHNICAL DIFFICULTY FOR APPLYING GENERAL OPTIMAL TRANSPORT/MARKOV THEORY TO LABEL SPACE.

Let $X, Y$ be uncertain and certain samples in pre-softmax feature space. And assume that we consider the distributional constraint on label-space (the space of $\sigma(X), \sigma(Y)$, where $\sigma$ denotes the soft-max function). This space is not proper to define the objective function such as (5). Because, all the samples in this label space is of the form $\sigma(X) = [a_1, a_2, \cdots, a_n]$ such that $\sum_{i=1}^{d} a_i = 1$, thus label-space is $d$-dimensional affine-simplex $U_d$ which is subset of Euclidean space $U_d \subset \mathbb{R}^d$. In this case, the definition of Wasserstein space in equation (4) is unacceptable while $d_E$ is not true metric on $U_d$. The Wasserstein space $\mathcal{P}_2(U_d)$ is merely investigated in the mathematical literature which makes unable to use all the technical details and assumptions, theories developed in the $\mathcal{P}_2(\mathbb{R}^d)$ which are theoretical ground of our work. But, if we look this problem slightly different point of view, for example, consider pre-softmax $\mathbb{R}^d, \mathcal{P}_2(\mathbb{R}^d)$ as our base space. In this case, all the technical issues/problems when we try to use OT tools in $\mathcal{P}_2(U_d)$ can be overcome/ignored. while softmax is non-parametric one-to-one function connecting pre-softmax feature space $\mathbb{R}^d$ to $U_d$, there exists a unique labels in $U_d$ as a mapped point of the manipulated uncertain samples. Even though our objects are defined on pre-softmax space, the theoretical analysis in Proposition 3 contains softmax function to evaluate the concentration inequality of proposed transformation $\mathcal{F}$ affecting in label-space $U_d$.

## D    MATHEMATICAL BACKGROUND

In this section, we introduce important definitions, notations, and propositions used in our proofs and the main paper.

### D.1    NOTATION

We denote $f_{\#}\mu$ as a push-forward of $\mu$ through $f$. $C_0^{\infty}(\mathbb{R}^d)$ denotes the set of $\infty$-class functions with compact support in $\mathbb{R}^d$. For the $L_p$-norm of the function $f$, we denote $\|f\|_{p,\nu} = (\int |f|^p d\nu)^{\frac{1}{p}}$. The Hessian matrix of the function $f$ is denoted as $\mathbf{Hess}[f] = [\partial_i \partial_j f]_{i,j}^d$. $\mathbf{Sym}_d^+$ denotes the space for semi-definite positive symmetric matrices of size $d \times d$. $\|f\|_{Lip}$ denotes the Lipschitz norm of the function $f$. For any matrix $A \in \mathbb{M}_d$, we let $\|A\|_{op}$ denote the operator norm of $A$.

### D.2    DIFFUSION-INVARIANCE AND HYPER-CONTRACTIVITY

**Definition 2.** *The Markov semigroup $(P_t)_{t\geq 0}$ in $\mathbb{R}^d$ acting on a function $f \in C_0^{\infty}$ is defined as follows:*

$$P_t f(x) = \int f(x') p_t(x, dx'), \tag{15}$$

*where $p_t(x, dx')$ is a transition kernel that is the probability measure for all $t \geq 0$.*

**Definition 3.** *(Diffusion Operator) Given a Markov semi-group $P_t$ at time $t$, the diffusion operator (i.e., infinitesimal generator) $\mathcal{L}$ of $P_t$ is defined as*

$$\mathcal{L}g(y) = \lim_{t \to 0} \frac{1}{t}(P_t g(y) - g(y)) = \sum_{i,j} \frac{\partial^2}{\partial y_i \partial y_j} B^{ij}(y)g(y) - \sum_i A^i(y)\frac{\partial}{\partial y_i}g(y), \tag{16}$$

*where $B$ and $A$ are matrix and vector-valued measurable functions, respectively. $B^{ij}$ denotes the $(i, j)$-th function of $B$ and $A^i$ denotes the $i$-th component function of $A$.*

**Definition 4.** *(Diffusion-invariant Measure) Given the diffusion operator $\mathcal{L}$, the probability measure $\mu$ is considered to be invariant measure to $\mathcal{L}$ when $\mathbb{E}_{X\sim\mu}[\mathcal{L}f(X)] = 0$ for any $f \in C_0^{\infty}$.*

**Lemma 1.** *(Infinitesimal generator for the multivariate Gaussian measure, Bolley & Gentil (2010).) The Gaussian measure $\mathcal{N}_{\nu} := \mathcal{N}(\mathbf{m}_{\nu}, \Sigma_{\nu})$ with a mean $\mathbf{m}_{\nu}$ and covariance $\Sigma_{\nu}$ is an invariant measure according to the following diffusion-operator $\mathcal{L}$:*

$$\mathcal{L}f(x) = \Sigma_{\nu}\mathbf{Hess}[f](x) - (x - \mathbf{m}_{\nu})^T \nabla f(x), \quad \forall f \in C_0^{\infty}(\mathbb{R}^d), \tag{17}$$

*where $B^{ij}(x) := [\Sigma_{\nu}]_{ij}$ is a constant function, and $A^i(x) := x^i - \mathbf{m}_{\nu}^i$.*

This generator serves as our main tool for the geometric analysis of the upper bound $\varepsilon$. In Section 4.1 in the main paper, we introduced an approximate upper-bound $\hat{K}_2(\mu)$ without any general description of the inequality involved. We now introduce the underlying mathematics for equation 12. Because our detour measure is Gaussian, there is a unique semi-group $P_t h$ called the *multidimensional Ornstein-Ulenbeck semi-group* that is invariant to $\mathcal{N}_\nu$. Specifically, $P_t$ is defined as follows:

$$P_s h(X) = \mathbb{E}_{Z \sim \mathcal{N}_\mathbf{I}} \left[ h \left( e^{-s} X + \sqrt{1 - e^{-2s}} (\Sigma_\nu^{\frac{1}{2}} Z + \mathbf{m}_\nu) \right) \right], \quad \forall h \in C_0^\infty. \tag{18}$$

The invariance property of $P_t$ relative to our detour measure is naturally induced by the following Proposition:

**Proposition 4.** *We define $C : \mathbb{R}^d \to \mathbb{R}^d$ and $C(X) = \mathbf{A}X + \mathbf{b}$ such that $\mathbf{A} \in \mathbf{Sym}_d^+, \mathbf{b} \in \mathbb{R}^d$, and select an arbitrary smooth $h \in C_0^\infty(\mathbb{R}^d)$. We then define the diffusion Markov semi-group $P_s h$ as follows:*

$$P_s h(X) = \mathbb{E}_{Z \sim \mathcal{N}} \left[ h \left( e^{-s} X + \sqrt{1 - e^{-2s}} C(Z) \right) \right]. \tag{19}$$

*Then, $\mathcal{N}(\mathbf{A^2}, \mathbf{b})$ is invariant with respect to $P_s$, meaning the following equality holds for every $h$ and $s \geq 0$:*

$$\int_{\mathbb{R}^d} [P_s h(X) - h(X)] d\mathcal{N}(\mathbf{A^2}, \mathbf{b})(X) = 0. \tag{20}$$

*Proof.* For simplicity, we denote $\mathcal{N}(\mathbf{A^2}, \mathbf{b}) := \mathcal{N}_C$.

$$\begin{aligned}
\int P_s h(X) d\mathcal{N}_C(X) &= \int \int h(e^{-s} X + \sqrt{1 - e^{-2s}} C(Z)) d\mathcal{N}_C(X) d\mathcal{N}(Z) \\
&= \int \int h \circ C(e^{-s} Z' + \sqrt{1 - e^{-2s}} Z) d\mathcal{N}(Z') d\mathcal{N}(Z).
\end{aligned} \tag{21}$$

The second equality holds because $C$ is linear in $\mathbb{R}^d$. Let $e^{-s} = \cos \theta$ and $e^{-2s} = \sin \theta$ for any $0 \leq \theta \leq 2\pi$. Then, we define $\phi$ as $\phi(Z', Z) = e^{-s} Z' + \sqrt{1 - e^{-2s}} Z = \cos(\theta) Z' + \sin(\theta) Z$, and $\pi(Z', Z) = Z$. Based on the rotation property of the standard Gaussian measure, one can induce the following equality.

$$(\mathcal{N} \otimes \mathcal{N}) \circ (C \circ \phi)^{-1} = ((\mathcal{N} \otimes \mathcal{N}) \circ \phi^{-1}) \circ C^{-1} = \mathcal{N} \circ C^{-1}. \tag{22}$$

However, we know that $d\mathcal{N}[C^{-1}(X)] = d\mathcal{N}_C(X) = \left( (2\pi)^d |\mathbf{A}^2| \right)^{-\frac{1}{2}} e^{-0.5(X - \mathbf{b})^T \mathbf{A}^{-2}(X - \mathbf{b})}$. By combining equation 21 and equation 22, one can derive the following result:

$$\begin{aligned}
\int h \circ C(e^{-s} Z' + \sqrt{1 - e^{-2s}} Z) d[\mathcal{N} \otimes \mathcal{N}] &= \int h(X) d \left[ (\mathcal{N} \otimes \mathcal{N}) \circ \phi^{-1} \circ C^{-1} \right](X) \\
&= \int h(X) d[\mathcal{N} \circ C^{-1}](X) = \int h(X) d\mathcal{N}[C^{-1}(X)] \\
&= \int h(X) d\mathcal{N}_C(X).
\end{aligned} \tag{23}$$

$\square$

Proposition 4 demonstrates the invariance property of the defined semi-group. If we set $A = \Sigma_\nu^{\frac{1}{2}}, \mathbf{b} = \mathbf{m}_\nu$, then we can recover equation 18.

We are now ready to define the approximation of $K_2(\mu)$ in terms of semi-group invariance. Specifically, for any real-valued smooth $h$, we define the following inequality:

$$\begin{aligned}
\hat{K}_2(\mu) = \mathbb{E}_{X \sim \mu}[\mathcal{L}h(X)] &= \lim_{s \to 0} \mathbb{E}_{X \sim \mu} \left[ \frac{1}{s} (P_s h(X) - h(X)) \right] \\
&= \lim_{s \to 0} \frac{1}{s} \mathbb{E}_{X, Z \sim \mathcal{N}_\mathbf{I}} \left[ h \left( e^{-s} X + \sqrt{1 - e^{-2s}} (\Sigma_\nu^{\frac{1}{2}} Z + \mathbf{m}_\nu) \right) - h(X) \right] \leq K_2(\mu).
\end{aligned} \tag{24}$$

This inequality holds if $h$ is selected to induce a supremum over the set $C_0^\infty$, where $\sup_h \hat{K}_2(\mu, h) = \sup_h \mathbb{E}_{X \sim \mu}[\mathcal{L}h(X)] = K_2(\mu)$. Although a more sophisticated design for the test function $h$ will induce a tighter upper bound for $\hat{K}_2$, we determined that the $L_2$-norm is generally sufficient.

**Definition 5.** *(Diffuseness of the probability measure) We define the integral operator $K_2$ : $\mathcal{W}_2(\mathbb{R}^d) \to \mathbb{R}^+$ as follows:*

$$K_2(\mu) = \sqrt{\sup_{f \in C_0^\infty} \int_{\mathbb{R}^d} |\mathcal{L}f(x)| \, d\mu(x)}. \tag{25}$$

According to Definition 4, we know that $\int \mathcal{L}f(X)d\mathcal{N}_\nu(X) = 0$ for any $f$. Based on this observation, it is intuitive that $K_2$ estimates how the probability measure $\nu$ is distorted in terms of diffusion invariance. While this measure takes a supremum over the function space $C_0^\infty$, it searches for a function that enables the estimation of maximal distortion. Because the value of $K_2$ is entirely dependent on the structure of $\mu$, $K_2$ can be considered as a constant for the sake of simplicity if the uncertain measure $\mu$ is fixed over one iteration of training.

**Definition 6.** *(Diffusion carré du champ) Let $f, g \in C_0^\infty(\mathbb{R}^d)$. Then, we define a bilinear form $\Gamma_c$ in $C_0^\infty(\mathbb{R}^d) \times C_0^\infty(\mathbb{R}^d)$ as*

$$\Gamma_e(f,g) = \frac{1}{2}[\mathcal{L}\Gamma_{e-1}(fg) - \Gamma_{e-1}(f\mathcal{L}g) - \Gamma_{e-1}(g\mathcal{L}f)], \quad e \geq 1. \tag{26}$$

We also denote $\Gamma(f) \equiv \Gamma(f, f)$. The bilinear form $\Gamma$ can be considered as a generalization of the integration by the parts formula, where $\int f\mathcal{L}g + \Gamma(f)d\mu = 0$ for the invariant measure $\mu$ of $\mathcal{L}$.

**Definition 7.** *(Curvature-Dimension condition, Ambrosio et al. (2015)) We can say that the infinitesimal generator $\mathcal{L}$ induces the $CD(\rho, \infty)$ curvature-dimension condition if it satisfies $\Gamma_1(f) \leq \rho\Gamma_2(f)$ for all $f \in C_0^\infty$.*

Because our diffusion operator generates a semi-group with respect to the Gibbs measure, the curvature-dimension condition can be calculated explicitly. Through simple calculations, the first-order $(c = 1)$ diffusion carré du champ can be induced as follows:

$$\Gamma_1(f) = \left([\nabla f]^T \Sigma_\nu \nabla f\right)^2. \tag{27}$$

Similarly, the second-order $(c = 2)$ diffusion carré du champ is calculated as follows:

$$\begin{aligned}
\Gamma_2(f) &= \frac{1}{2}\left[\mathcal{L}\left(\Gamma_1(f^2)\right) - 2\Gamma_1\left(f, \mathcal{L}(f)\right)\right] \\
&= \mathbf{Tr}\left(\left[\Sigma_\nu \nabla^2 f\right]^2\right) + \left([\nabla f]^T \Sigma_\nu \nabla f\right)^2 = \mathbf{Tr}\left(\left[\Sigma_\nu \nabla^2 f\right]^2\right) + \Gamma_1(f),
\end{aligned} \tag{28}$$

for an arbitrary $f \in C_0^\infty(\mathbb{R}^d)$. While $\mathbf{Tr}\left(\left[\Sigma\nabla^2 f\right]^2\right)$ is non-negative, we can infer that $\Gamma_1 \leq \Gamma_2$. In this case, the diffusion operator $\mathcal{L}$ defined in Lemma 1 induces the $CD(\rho = 1, \infty)$ curvature-dimension condition. For the other diffusion operators, please refer to Bolley & Gentil (2010).

**Proposition 5.** *(Decay of Fisher information along a Markov semigroup, Bakry et al. (2013).) If we assume the curvature-dimension condition $CD(\rho, \infty)$, then $I(\mu_t|\mathcal{N}_\nu) \leq e^{-2\rho t}I(\mu|\mathcal{N}_\nu)$.*

The exponential decay of the Fisher information in Proposition 5 is a core property of the exponential decay of the Wasserstein distance, which will be used in the proof of Proposition 2.

### D.3 FOKKER-PLANK EQUATION, SDE

**Definition 8.** *(Over-damped Langevin Dynamics) We have*

$$dX_t = -\nabla\phi(X_t; \mathbf{m}_\nu)dt + \sqrt{2\tau^{-1}\Sigma_\nu}dW_t, \tag{29}$$

*where $\phi(X_t; \mathbf{m}_\nu) = \frac{\tau}{2}d^2(X_t, \mathbf{m}_\nu)$, $W_t$ denotes Brownian motion, and $d$ denotes Euclidean distance. The particle $X_t$ is distributed in $X_t \sim p_t$. The probability density $\lim_{t\to\infty} p(x, t)$ with respect to $X_\infty$ converges to the Gaussian density $X_\infty = \sqrt{\Sigma_\nu}(Z + \mathbf{m}_\nu) \sim p_\infty(x) = q(x) \propto e^{-d(x,\mathbf{m}_\nu)^T \Sigma_\nu^{-1} d(x,\mathbf{m}_\nu)}$.*

In classical SDE literature, it is stated that $\mathbb{E}\left[\sup_{0 \leq t \leq T}\left|\hat{X}_t - X_t\right|\right] \leq G(N_\varrho)^{-\frac{1}{2}}$, where $G(T)$ is some constant that depends only on $T$ and $\hat{X}$ denotes the true solution of the SDE in equation 29. While the number of uncertain samples is greater than $N_\varrho > 40$, our method exhibits acceptable convergence.

### D.4 GAUSSIAN WASSERSTEIN SUBSPACES

It is known that the space of non-degenerate Gaussian measures (*i.e.*, covariance matrices are positive-definite) forms a subspace in the 2-Wasserstein space denoted as $\mathcal{W}_{2,g} \cong \mathbf{Sym}_d^+ \times \mathbb{R}^d$. Because the 2-Wasserstein space can be considered as a Riemannian manifold equipped with Riemannian metrics Villani (2008), $\mathcal{W}_{2,g}$ can be endowed with a Riemannian structure that also induces the Wasserstein metric (McCann (1997)). In the Riemannian sub-manifold of Gaussian measures, the geodesic between two points $\gamma(0) = \mathcal{N}_A$ and $\gamma(1) = \mathcal{N}_B$ is defined as follows Malagò et al. (2018):

$$\gamma(\alpha) = \mathcal{N}_t = \mathcal{N}(\mathbf{m}(\alpha), \Sigma(\alpha)), \tag{30}$$

where $\mathbf{m}(\alpha) = (1 - \alpha)\mathbf{m}_A + \alpha\mathbf{m}_B$ and $\Sigma(\alpha) = [(1 - \alpha)\mathbf{I} + \alpha\mathcal{T}] \Sigma_A [(1 - \alpha)\mathbf{I} + \alpha\mathcal{T}]$, where $\mathcal{T}\Sigma_A\mathcal{T} = \Sigma_B$. In Section 3.2, we set $(\mathbf{m}_A, \Sigma_A) \to (\mathbf{m}_\nu, \Sigma_\nu)$ and $(\mathbf{m}_B, \Sigma_B) \to (\mathbf{m}_{\xi_k}, \Sigma_{\xi_k})$. Regardless of how $\nu$ is updated, the statistical information regarding the current certain measure $\xi_k$ is considered in the detour Gaussian measure, which yields a much smoother geometric constraint on $\mu$.

## E PROOFS

**Proposition 6.** *Let $\Gamma(\mu, \nu)$ be a set of couplings between $\mu$ and $\nu$, and assume that the noisy label $\hat{r}$ is independent of $X$. For functional $\mathcal{J}[\mu] = \mathbb{E}_{\mu \sim X} l(X; \hat{r})$, we define $D(\mu, \nu)$ as:*

$$D(\mu, \nu) = \inf_{\gamma \in \Gamma(\mu, \nu)} |\mathcal{J}[\mu] - \mathcal{J}[\nu]|, \tag{31}$$

*where $D : \mathcal{P}_2 \times \mathcal{P}_2 \to \mathbb{R}$. Then, $D$ is the metric defined on $\mathcal{P}_2$, which is weaker than the Wasserstein metric, where $D(\mu, \nu) \leq \alpha\mathcal{W}_2(\mu, \nu)$ for $\alpha = c_0^{-1}\hat{r} + c_1^{-1}(1 - \hat{r})$ and some constants $c_0, c_1 > 0$.*

*Proof.*

$$
\begin{aligned}
|\mathcal{J}[\nu] - \mathcal{J}[\mu]| &= |\mathbb{E}_\mu[l(X; \hat{r})] - \mathbb{E}_\nu[l(Z; \hat{r})]| \\
&= |\mathbb{E}_{\mu \otimes \nu} [\hat{r} (\log \sigma(X) - \log \sigma(Z)) - (1 - \hat{r}) (\log(1 - \sigma(X)) - \log(1 - \sigma(Z)))]| \\
&\leq \mathbb{E} |\hat{r}\mathbb{E}_{\mu \otimes \nu} [\log \sigma(X) - \log \sigma(Z)]| + \mathbb{E} |(1 - \hat{r})\mathbb{E}_{\mu \otimes \nu} [\log(1 - \sigma(X)) - \log(1 - \sigma(Z))]| \\
&\leq \mathbb{E}\hat{r}\mathbb{E}_{\mu \otimes \nu} |\log \sigma(X) - \log \sigma(Z)| + \mathbb{E}(1 - \hat{r})\mathbb{E}_{\mu \otimes \nu} |\log(1 - \sigma(X)) - \log(1 - \sigma(Z))| \\
&\leq c_0^{-1}\mathbb{E}(\hat{r})\mathbb{E}_{\mu \otimes \nu}|X - Z| + c_1^{-1}\mathbb{E}(1 - \hat{r})\mathbb{E}_{\mu \otimes \nu}|Z - X| \\
&= \mathbb{E}[c_0^{-1}\hat{r} + c_1^{-1}(1 - \hat{r})]\mathbb{E}_{\mu \otimes \nu}|X - Z|
\end{aligned}
\tag{32}
$$

By taking the infimum of the aforementioned inequality with set of couplings $\gamma(\mu, \nu)$, we obtain the following inequality:

$$
\begin{aligned}
D(\nu, \mu) = \inf_{\gamma(\mu, \nu)} |\mathcal{J}[\nu] - \mathcal{J}[\mu]| &\leq \mathbb{E}[c_0^{-1}Y + c_1^{-1}(1 - Y)] \inf_{\gamma(\mu, \nu)} \mathbb{E}_\gamma|X - Z| \\
&= \mathbb{E}[c_0^{-1}Y + c_1^{-1}(1 - Y)]\mathcal{W}_1(\mu, \nu) \\
&\leq \mathbb{E}[c_0^{-1}Y + c_1^{-1}(1 - Y)]\mathcal{W}_2(\mu, \nu),
\end{aligned}
\tag{33}
$$

which completes the proof. □

Proposition 6 follows from the Lipschitzness of the functional $\mathcal{J}$, where $D$ searches for the best coupling to derive the minimal loss difference between two probability measures. This proposition indicates that $\inf |\mathcal{J}[\nu] - \mathcal{J}[\mathcal{F}\mu]|$ is bounded by the *Wasserstein distance*, which justifies our geometric constraint presented in equation 4. It should be noted that the prior assumption regarding noisy labels is essential for Lipschitzness.

**Proposition 7.** *Let $\mathcal{F} : \mathbb{R}^+ \times \mathcal{P}_2$ be a functional on probability measures such that $\mathcal{F}[t, \mu] = \mu_t$, where $d\mu_t = p_t d\mathcal{N}_\nu, d\mathcal{N}_\nu = dq_t dx$, and let $\mu_t$ be a solution of the continuity equation in the 2-Wasserstein space defined as follows:*

$$\partial_t \mu_t = \nabla \cdot (\mu_t \nabla \Phi_t), \tag{34}$$

*which is represented as $\partial_t p(t, x) = \nabla \cdot (p(t, x)\nabla \log q(t, x))$ in a distributional sense. Then, the functional $\mathcal{F}_t[\cdot] = \mathcal{F}[t, \cdot]$ is defined unique and normalizes $\mu$ onto $\mathbb{B}_{\mathcal{W}_2}(\mathcal{N}_\nu, e^{-t}K_2(\mu))$, where $K_2(\mu) \leq \infty$ is an integral operator in Definition 5 with respect to $\mu$.*

*Proof.* We assume that the probability measure $\mu_t$ is absolutely continuous with respect to the detour Gaussian measure $\mathcal{N}(\mathbf{m}_\nu, \Sigma_\nu) = \mathcal{N}_\nu$, $\mu_t \ll \mathcal{N}_\nu$. In this case, according to the Radon-Nikodym theorem, there is a corresponding unique probability density $q(t, x) = q_t(x) \in C_0^\infty$ such that $d\mu_t = q_t d\mathcal{N}_\nu$.

**Lemma 2.** *(**WI-inequality**, Otto & Villani (2000)) If the stationary state of $\mu_t$ with respect to $P_t$ satisfies $\lim_{t \to \infty} \mathbb{E}_\mu[P_t f] = 0$ for any $f \in C_0^\infty$, then the following inequality holds:*

$$\frac{d}{dt_+} \mathcal{W}_2(\mu, \mu_t) \leq \sqrt{I(\mu_t | \mathcal{N}_\nu)}. \tag{35}$$

By integrating both sides of the inequality in Lemma 2 with respect to $t \in (0, \infty)$, the following inequality can be obtained:

$$\mathcal{W}_2(\mu_t, \mathcal{N}_\nu) = \int_0^\infty \frac{d}{dt_+} \mathcal{W}_2(\mu_t, \mathcal{N}_\nu) dt \leq \int_0^\infty \sqrt{I(\mu_t | \mathcal{N}_\nu)} dt. \tag{36}$$

In the aforementioned inequality, we replace the Fisher information with the diffusion generator $\mathcal{L}$ as follows:

$$\mathcal{W}_2(\mu, \mathcal{N}_\nu) \leq \int_0^\infty \sqrt{I(\mu_t | \mathcal{N}_\nu)} dt$$
$$= \int_0^\infty \sqrt{\int [P_t q]^{-1} \Gamma(P_t q) d\mathcal{N}_\nu} dt = \int_0^\infty \sqrt{\int \mathcal{L}(-\log P_t q) d\mu_t} dt. \tag{37}$$

The second equality above is derived by leveraging the properties of the bilinear operator $\Gamma$ (Bakry et al. (2013); Villani (2008)) with respect to the diffusion operator $\mathcal{L}$, which is defined as follows:

$$\int [P_t q]^{-1} \Gamma(P_t q) d\mathcal{N}_\nu = -\int \mathcal{L}(\log P_t q) q_t d\mathcal{N}_\nu = \int \mathcal{L}(-\log P_t q) d\mu_t \geq 0. \tag{38}$$

For simplicity, we denote $|g| = g^+$ for any $g \in C_0^\infty$. According to Proposition 5, we can relate $\mathcal{F}_t \mu = \mu_t$ to its initial term $\mu = \mu_{t=0}$ as follows:

$$\int_0^\infty \sqrt{\int \mathcal{L}(-\log P_t q)(X) d[\mathcal{F}_t \mu](X)} dt \leq \int_0^\infty \sqrt{e^{-2\rho t} \int \mathcal{L}(-\log P_{t=0} q)(X) d\mu(X)} dt$$
$$\leq \int_0^\infty \sqrt{e^{-2\rho t} \sup_{g \in C_0^\infty} \int \mathcal{L}^+ g(Z) q d\mathcal{N}_\nu(Z)} dt \tag{39}$$
$$= \int_0^\infty \sqrt{e^{-2\rho t}} dt \sqrt{\sup_{g \in C_0^\infty} \int \mathcal{L}^+ g(X) d\mu(X)}$$
$$= \rho^{-1} K_2(\mu).$$

The second inequality is naturally induced, because the proposed objective function is defined to select the maximum elements over the set of functions $g \in C_0^\infty$ and $\mathcal{L}g \leq \mathcal{L}^+ g$. If the integral interval is set to $(0, s)$, then we can induce $\mathcal{W}_2(\mu, \mathcal{F}_t \mu) \leq \frac{1}{\rho}(1 - e^{-s}) K_2(\mu)$. Our diffusion-operator induces $\rho = 1$, which completes the proof. $\square$

**Proposition 8.** *There is a scalar $0 < \beta < \infty$ dependent on $\nu$ such that the following inequality holds:*

$$\mathcal{W}_2(\nu, \mathcal{F}_t \mu) \leq \left[ \sqrt{d\beta \lambda_{max}(\Sigma_\nu)} + \|\mathbb{E}_\nu Y\|_2 \right] \vee \left[ e^{-t} K_2(\mu) + K_2(\nu) \right]. \tag{40}$$

As a motivation for setting a detour measure to $\mathcal{N}_\nu$, we mentioned the natural property of the non-collapsing Wasserstein distance of $\mathcal{W}_2(\nu, \mathcal{N}_\nu) \neq 0$. However, it is unclear from a geometric perspective exactly how the upper bound (*i.e.*, $\mathcal{W}_2(\nu, \mathcal{N}_\nu) \leq ?$) can be induced based on the intrinsic statistics term (*i.e.*, $d_1$ in Fig.1). Specifically, in the situation where the covariance matrices of $\nu$ and $\mathcal{N}_\nu$ are identical, it is difficult to determine a theoretical upper bound without additional tools. The first part of this proof focuses on resolving this important issue. The second part of the proof is naturally induced by Proposition 1. Please note that in the following proposition, parameter for Wasserstein moving average is set to $\alpha = 0$ for clarity.

*Proof.* Before proceeding with the first part of the proof, we define a constant $\beta$ as follows:

$$\beta = \sup_{1 \le j \le d} \int_0^1 \frac{1}{s} \mathbb{E}_{Y_s} v_{s,j}^2(Y_s) ds. \tag{41}$$

If we assume a mild condition such that $\min_{s,j} \inf_{1 \le j \le d} O(v_{s,j}) \ge O(\sqrt{s})$, then the integral term in $\beta$ is finite and well-defined. This value will directly yield the upper bound of the Kullback–Leibler (KL) divergence of $\nu$. First, we introduce the following inequality.

**Lemma 3.** *(de Bruijn's identity, Johnson & Suhov (2001); Nourdin et al. (2014)) We let $Y \sim \nu$, $Z \sim \mathcal{N}(\mathbf{0}, \mathbf{I})$ denote a standard Gaussian random variable, and let define $Y_s = \sqrt{s}Y + \sqrt{1-s}\Sigma_\nu^{\frac{1}{2}} Z$ with the score function defined as $v_s(x) = \nabla \log p_s(x)$ with respect to the random variable $Y_s$. Then, the following equality holds:*

$$\mathbf{KL}(\nu | \mathcal{N}(\mathbf{0}, \Sigma_\nu)) = \int_0^1 \mathbf{Tr}\left( \frac{1}{2s} \Sigma_\nu \mathbb{E}_{p_s \sim Y_s}[v_s(Y_s)v_s(Y_s)^T] \right) ds. \tag{42}$$

From equation 42, we can derive the relations between KL-divergence and the constant $\beta$ defined earlier.

$$\int_0^1 \frac{1}{2s} \mathbf{Tr}\left( \Sigma_\nu \mathbb{E}_x[v_s(Y_s)v_s(Y_s)^T]) \right) ds \le \int_0^1 \frac{1}{2s} \mathbf{Tr}\left( \Sigma_\nu \mathbb{E}_x[v_{s,i}v_{s,j}]_{i,j}^d) \right) ds$$

$$\le \int_0^1 \frac{1}{2} \lambda_{max}(\Sigma_\nu) \sum_{j=1}^d \mathbb{E}\left[ \frac{v_{s,j}^2(Y_s)}{s} \right] ds \le \frac{1}{2}\lambda_{max} \int_0^1 \sum_{j=1}^d \beta ds = \frac{1}{2}\lambda_{max}(\Sigma_\nu)d\beta. \tag{43}$$

The second inequality holds based on the following element property of symmetric positive-definite matrices:

$$\mathbf{Tr}(AB) \le \|A\|_{op} \mathbf{Tr}(B) = \lambda_{max}(A)\mathbf{Tr}(B), \quad \forall A, B \in \mathbf{Sym}_d^+. \tag{44}$$

It should be noted that because the distribution of $\nu$ is compactly supported (*i.e.*, $\mathbf{supp}(q)$ is compact), the maximum eigenvalue of the covariance $\Sigma_\nu$ is finite. The other relations are induced by the aforementioned definition. Next, we relate the KL-divergence and 2-Wasserstein distance naturally.

**Definition 9.** *(Talagrand inequality for Gaussian measures, Otto & Villani (2000)) For any non-degenerate Gaussian measure $\mathcal{N}$ with a mean 0, the following inequality is satisfied:*

$$\mathcal{W}_2(\nu, \mathcal{N}) \le \sqrt{2\mathbf{KL}(\nu|\mathcal{N})}, \quad \forall \nu \in \mathcal{P}_2(\mathbb{R}^d). \tag{45}$$

By combining Definition 9 and equation 43, we can derive the following expression:

$$\mathcal{W}_2(\nu, \mathcal{N}(0, \Sigma_\nu)) \le \sqrt{2\mathbf{KL}(\nu|\mathcal{N}(0, \Sigma_\nu))} \le \sqrt{d\beta\lambda_{max}(\Sigma_\nu)} < \infty. \tag{46}$$

According to the triangle inequality for the 2-Wasserstein distance, we obtain:

$$\mathcal{W}_2(\nu, \mathcal{N}(\mathbf{m}_\nu, \Sigma_\nu)) \le \mathcal{W}_2(\nu, \mathcal{N}(0, \Sigma_\nu)) + \mathcal{W}_2(\mathcal{N}(\mathbf{m}_\nu, \Sigma_\nu), \mathcal{N}(0, \Sigma_\nu)) \tag{47}$$

In Appendix C.3, we investigated that the geodesic distance between two Gaussian measures having the same covariance is equivalent to the Euclidean distance between two means. Therefore, we can obtain the following equality:

$$\mathcal{W}_2(\mathcal{N}(\mathbf{m}_\nu, \Sigma_\nu), \mathcal{N}(0, \Sigma_\nu)) = \mathcal{W}_2(\iota_\#^{\mathbf{m}_\nu}[\mathcal{N}(0, \Sigma_\nu)], \mathcal{N}(0, \Sigma_\nu))$$
$$= \|\mathbf{m}_\nu - 0\|_2 = \|\mathbb{E}_\nu Y\|_2, \tag{48}$$

where $\iota^{\mathbf{a}}(X) = X + \mathbf{a}$ for any vector $\mathbf{a} \in \mathbf{supp}(q)$. Now, by adding the two inequalities defined earlier, we can obtain

$$\mathcal{W}_2(\nu, \mathcal{N}(\mathbf{m}_\nu, \Sigma_\nu)) \le \|\mathbb{E}_\nu Y\|_2 + \sqrt{d\beta\lambda_{max}(\Sigma_\nu)}, \tag{49}$$

where it is easily shown that the upper-bound is only dependent on the statistical structure of $\nu$. Specifically, the term $\|\mathbb{E}_\nu Y\|_2$ represents the center of mass for a density of $\nu$ and $\sqrt{d\beta\lambda_{max}(\Sigma_\nu)}$ is related to the covariance structure of $\nu$.

By applying Proposition 8 to both $\mathcal{F}_t\mu$ and $\nu$, we can easily recover equation 5 as follows:

$$
\begin{aligned}
\mathcal{W}_2(\nu, \mathcal{F}_t\mu) \leq \varepsilon &= \mathcal{W}_2(\nu, \mathcal{N}(\mathbf{m}_\nu, \Sigma_\nu)) + \mathcal{W}_2(\mathcal{N}(\mathbf{m}_\nu, \Sigma_\nu), \mathcal{F}_t\mu) \\
&\leq \left( \left[ \|\mathbb{E}_\nu Y\|_2 + \sqrt{d\beta\lambda_{max}(\Sigma_\nu)} \right] \wedge K_2(\nu) \right) + e^{-t}K_2(\mu) \\
&\leq \left[ \sqrt{d\beta\lambda_{max}(\Sigma_\nu)} + \|\mathbb{E}_\nu Y\|_2 \right] \vee \left[ e^{-t}K_2(\mu) + K_2(\nu) \right].
\end{aligned}
\tag{50}
$$

The second inequality is easily obtained as $(a \wedge b) + c \leq a \vee (b + c)$ for any $a, b, c \geq 0$, which completes the proof. $\qquad\square$

**Proposition 9.** *(Concentration inequality for uncertain measures). Assume that there are some constants $s^\star \in [\frac{1}{\eta}, \infty), \eta \geq 0$ such that the following inequality is satisfied:*

$$
\mathbb{E}_{\mathcal{F}_{s^\star}\mu}[f^2] - [\mathbb{E}_{\mathcal{F}_{s^\star}\mu}[f]]^2 \leq (1+\eta)\mathbb{E}_{\mathcal{F}_{s^\star}\mu}[\mathbf{A}\nabla f^T\nabla f],
\tag{51}
$$

*for $\mathbf{A} \in \mathbf{Sym}_d^+$, $D(\mathbf{A}, \Sigma_\nu) \leq a\eta$ for some $a > 0$, and for any metric $D$ defined on $\mathbf{Sym}_d^+$. In this case, there is a $\delta$ such that the following probability inequality for an uncertain measure is induced:*

$$
\mathcal{F}_{s^\star}\mu\left( |\sigma - \mathbb{E}_\nu[\sigma]| \geq \delta \right) \leq 6e^{-\frac{\sqrt{2}\delta^{\frac{3}{2}}}{K_2}},
\tag{52}
$$

*where $\kappa$ denotes the Lipschitz constant of $\sigma$.*

*Proof.* Before proceeding with the main proof, we first prove the existence of $s^\star$. The limit of the interval with respect to $\eta$ converges to a singleton $\{\infty\}$ as $I = \lim_{\eta\to 0}[\frac{1}{\eta}, \infty)$. In this case, equation 51 is the same as the Poincaré inequality for a Gaussian measure $\mathcal{N}_\nu$, which can be written as

$$
\begin{aligned}
\lim_{\eta\to 0} \mathbb{E}_{\mathcal{F}_{s^\star}\mu}[f^2] - [\mathbb{E}_{\mathcal{F}_{s^\star}\mu}[f]]^2 &\leq \lim_{\eta\to 0}(1+\eta)\mathbb{E}_{\mathcal{F}_{s^\star}\mu}[\mathbf{A}\nabla f^T\nabla f] \\
&= \mathbb{E}_{\mathcal{F}_{s^\star}\mu}[\Sigma_\nu\nabla f^T\nabla f].
\end{aligned}
\tag{53}
$$

While the Poincaré inequality in equation 53 is uniquely defined, we can find at least one value $s^\star$ satisfying equation 51. Let $X(t, w) = X_t(w)$ denote the stochastic process with respect to $q_t(x)$ defined in the proof of Proposition 2. Additionally, let $c = \mathbb{E}_\nu[\sigma] - \mathbb{E}_{\mathcal{F}_{s^\star}\mu}[\sigma]$. Then, we can obtain the following inequality:

$$
\begin{aligned}
c = \mathbb{E}_\nu[\sigma] - \mathbb{E}_{\mathcal{F}_{s^\star}\mu}[\sigma] &= \kappa\left( \mathbb{E}_\nu\left[\frac{\sigma}{\kappa}\right] - \mathbb{E}_{\mathcal{F}_{s^\star}\mu}\left[\frac{\sigma}{\kappa}\right] \right) \leq \kappa \sup_{g\in\mathbf{Lip}_1} (\mathbb{E}_\nu g - \mathbb{E}_{\mathcal{F}_{s^\star}\mu}g) \\
&\leq \kappa\mathcal{W}_1(\mathcal{F}_{s^\star}\mu, \nu) \leq \kappa\mathcal{W}_2(\mathcal{F}_{s^\star}\mu, \nu) \leq \frac{\kappa K_2(\mu)}{1+\eta}.
\end{aligned}
\tag{54}
$$

The first inequality is induced by the assumption regarding the $\kappa$-Lipschitzness of the function $\sigma$ and the second inequality is induced by the Kantorovich-Rubinstein theorem. The third inequality is natural because $\mathcal{W}_a(\cdot, \cdot) \leq \mathcal{W}_b(\cdot, \cdot)$ for any $1 \leq a \leq b < \infty$. because equation 51 is equivalent to the Poincaré inequality for the measure $\mathcal{F}_{s^\star}\mu$, it satisfies the Bakry-emery curvature-dimension condition $CD(1+\eta, \infty)$. Thus, as shown in the proof of Proposition 2 (*i.e.*, equation 39), the last inequality is induced. Additionally, based on the concentration inequality of $\mathcal{F}_{s^\star}\mu$ [Proposition 4.4.2 Bakry et al. (2013)], we can derive the following probability inequality:

$$
\mathcal{F}_{s^\star}\mu\left[ \sigma(X_{s^\star}(w)) \geq \mathbb{E}_{\mathcal{F}_{s^\star}\mu}[\sigma] + \delta \right] \leq 3e^{-\frac{\delta}{\sqrt{1+\eta}\kappa}},
\tag{55}
$$

where the Poincaré constant for $\mathcal{F}_{s^\star}\mu$ is naturally $1+\eta$ and $\|\sigma\|_{Lip} = \kappa$. Next, we will derive the desired form from equation 55. First, we introduce the following inequality.

$$
\sigma(X_{s^\star}) \geq \mathbb{E}_{\mathcal{F}_{s^\star}\mu}[\sigma] + \delta \geq \mathbb{E}_\nu[\sigma] + \delta - \frac{\kappa}{1+\eta}K_2
\tag{56}
$$

The last inequality is directly induced by equation 54 because $-c \geq -\frac{\kappa}{1+\eta}K_2$. While $\eta, \kappa$, and $K_2$ are constants with respect to $w$, the following set inclusion can be obtained naturally:

$$
\mathcal{S}_1 = \{w : \sigma(X_{s^\star}(w)) \geq \mathbb{E}_{\mathcal{F}_{s^\star}\mu}[\sigma] + \delta\} \supseteq \{w : \sigma(X_{s^\star}(w)) \geq \mathbb{E}_\nu[\sigma] + \delta - \frac{\kappa}{1+\eta}K_2\} = \mathcal{S}_2.
\tag{57}
$$

For the modified version of the original probability inequality, we take probability measure $\mathcal{F}_{s^\star}\mu[\cdot]$ for the sets $\mathcal{S}_1, \mathcal{S}_2$, which is defined as

$$
\begin{aligned}
3e^{-\frac{\delta}{\sqrt{1+\eta\kappa}}} &\geq \mathcal{F}_{s^\star}\mu\left(\{w : \sigma(X_{s^\star}(w)) \geq \mathbb{E}_{\mathcal{F}_{s^\star}\mu}[\sigma] + \delta\}\right) \\
&\geq \mathcal{F}_{s^\star}\mu\left(\{w : \sigma(X_{s^\star}(w)) \geq \mathbb{E}_\nu[\sigma] + \delta - \frac{\kappa}{1+\eta}K_2\}\right).
\end{aligned}
\tag{58}
$$

The concentration inequality around $\mathbb{E}_\nu[\sigma]$ is obtained by combining the inequalities induced by $\sigma$ and $-\sigma$ as follows:

$$
\begin{aligned}
&\frac{1}{2}\mathcal{F}_{s^\star}\mu\left(\bigcup_{h\in\{\sigma,-\sigma\}}\{w : h(X_{s^\star}(w)) - \mathbb{E}_\nu[h] \geq \pm\left(\delta - \frac{\kappa}{1+\eta}K_2\right)\}\right) \\
&= \mathcal{F}_{s^\star}\mu\left(\{w : |\sigma(X_{s^\star}(w)) - \mathbb{E}_\nu[\sigma]| \geq \delta - \frac{\kappa}{1+\eta}K_2\}\right) \leq 6e^{-\frac{\delta}{\sqrt{1+\eta\kappa}}}.
\end{aligned}
\tag{59}
$$

The inequality in equation 59 is the general form containing the relation between the upper bound of the probability and $(\eta, \kappa, K_2)$. While this form is quite complicated and highly technical, we choose not to present all the detailed expressions of equation 59 in the main paper. Rather than that, we re-write it in a much simplified form for clarity. Specifically, by setting $\kappa K_2/(1+\eta) = 0.5\delta$ and rescaling $\delta$ to $2\delta$, the aforementioned inequality in equation 59 can be converted into the following simpler form:

$$
\mathcal{F}_{s^\star}\mu\left(\{w : |\sigma(X_{s^\star}(w)) - \mathbb{E}_\nu[l]| \geq \delta\right) \leq 6e^{-\frac{\sqrt{2}\delta^{\frac{3}{2}}}{\kappa K_2}}.
\tag{60}
$$

$\square$

Finally, if we set $\sigma = \textbf{Softmax}$, then the Lipschitz constant is induced as $\kappa = 1$. This proof is completed by setting $s^\star := T$.

