# OpenReview forum: "Wasserstein Distributional Normalization : Nonparametric Stochastic Modeling for Handling Noisy Labels"
_ICLR.cc/2021/Conference — Reject_

### Official Review · AnonReviewer1 · 2020-10-26
**Related work, clarity, and rationale.**

**Rating:** 5
**Confidence:** 3

**Review:**

Note: initial score was 4; updated to 5 after taking a brief look at the experimental sections; will check again later to take a closer look at the updated "motivation" part.

---

__0. disclaimer.__
It is possible that I failed to understand a significant portion of this manuscript; I had a very hard time trying to understand the notations and the writing overall. Please correct me if I am wrong.

__1. paper summary.__
The paper aims to give a better way of handling the noisy label data, by providing a way to better utilize the information given by _uncertain_ (or high-loss) samples, which are often simply filtered out in previous works. The proposed method is as follows: (1) select the "lowest-loss mini-batch" among the _certain samples_ (i.e., low-loss) $Y$, with some empirical measure $\nu$. (2) Construct an approximate optimal transport map $\mathcal{F}$ from the measure of uncertain samples $\mu$ to a Gaussian measure $\mathcal{N}(\mathbf{m}_\nu,\Sigma_\nu)$ in the feature space, having the same first and second moments as $\nu$. (3) Minimize the sum of losses on the certain samples and transported uncertain samples. Authors also introduce several techniques to circumvent

__2. review summary.__
The proposed method seems to be advantageous over the considered baselines, but I am not sure if the method would outperform other baselines as well. Also, the clarity of the manuscript is not quite good (in my opinion).

__3. missing baselines.__
A wide range of algorithms has been proposed recently to make use of the estimated-to-be-mislabeled subset of samples (or uncertain sample, as authors put it) data. For instance, the ICLR 2020 paper by Li et al. (["DivideMix: Learning with noisy labels as semi-supervised learning"](https://openreview.net/pdf?id=HJgExaVtwr)) proposes to drop the labels from the uncertain samples and use semi-supervised learning approaches. The paper also refers to several related approaches, which could/should be compared or discussed against the optimal transport based method proposed by the authors. Other popular baselines would be [INCV](https://arxiv.org/abs/1905.05040) and the beta-mixture modeling by [Arazo et al.](https://arxiv.org/pdf/1904.11238.pdf), which are slightly different in spirit. A recent [NeurIPS 2020 paper](https://arxiv.org/pdf/2001.10528.pdf) (which should be considered as a concurrent work) is worth discussing, for the benefit of the readers.

I strongly recommend adding (at least) an empirical comparison to the Li et al., as it shares a larger goal (in the sense that utilizing the information from mislabeled data) yet implemented with a different philosophy. Understanding when/how one approach outperforms the other should be fantastic.

__4. clarity and (potential) errors.__
Some parts of the manuscript was not clear enough to me, or perhaps is written in an overly convoluted manner. Clarifying the following bits may help the readers (and me) follow the content more easily.
- Eq. (2) follows the description "Specifically, with a probability $1-\delta$, the generalization bound for clean datasets is defined as follows:". However, I am not sure whether it is a "definition of the generalization bound." Rather, it should be one of many generalization bounds that one can prove.
- Eq. (3) contains the expression $\mathbb{E}_{X\sim \mu,\hat{r}|T} [l(X;\hat{r})]$. I am not sure if the definition of $T$ appears in the main text. I guess that it is the label transition matrix? Also, on which space is the minimum taken over? Is it over any distribution $\mu$ on $\mathbb{R}^d$, or is it more like over any distribution that can be generated by partitioning $\mathbb{P}$? Any vagueness must be removed, as the authors also provide theoretical results.
- In definition 1, what distance measure $d(\cdot,\cdot)$ are authors using? (also, using same $d$ for distance and dimensions add unnecessary difficulty). If the authors are not using more than two different notions of distances, why don't you just write it directly, like $\mathbb{E}_{\mu}\lVert x_0-x\rVert_2^2$?
- In Eq. (5), again it is not clear over what the minimum is being taken over. Is it over all partitioned-distribution $\mu, \xi$, or also over all possible transport maps $\mathcal{F}$?

__5. Rationale__
I am curious what the general intuition behind the proposed approach is. Why is the "minimum transportation cost" map $\mathcal{F}$ a sensible method to transport the high-loss samples? Are we making any implicit assumptions on the nature of label-flipping operations, or perhaps the learning dynamics itself?

__minor suggestions.__
- The title can be more informative. I am not sure if the current title "Wasserstein distributional normalization" gives any information about the task under consideration, or
- The notation $\vartheta$ looks like the model parameter at a first glance. How about changing it to other greek letters?
- In proposition 1, there should be a typo: $\mathcal{F}: \mathbb{R}^+ \times \mathcal{P}_2$. The mapping $\mathcal{F}$ maps to where?

---

> ### Author Response · Authors · 2020-11-12
> **Response to Reviewer#1, (1/3)**
>
> $\textbf{ii-1)}$ Eq. (2) follows the description "Specifically, with a probability , the generalization bound for clean datasets is defined as follows:". However, I am not sure whether it is a "definition of the generalization bound." Rather, it should be one of many generalization bounds that one can prove.
>
> $\rightarrow$ As mentioned by Reviewer#1 , this is 'not' a unique representation of the generalization bound. This part is modified in the revised paper.
>
> $------------------------------------------$
>
> $\textbf{ii-2)}$ Eq. (2) contains $\mathbb{E}_{X \sim \mu, \hat{r}}[l(X ; \hat{r})]$ the expression. I am not sure if the definition of $T$ appears in the main text. I guess that it is the label transition matrix?
>
> $\rightarrow$ Yes. $T$ stands for the label transition matrix. In the revised version, the definition was clarified and the notation $T$ was replaced by $Q$. In Section 5.1, the mathematical description of noise transition matrix was defined.
>
> $------------------------------------------$
>
> $\textbf{ii-3)}$ On which space is the minimum taken over? Is it over any distribution $\mu$ on $\mathbb{R}^d$, or is it more like over any distribution that can be generated by partitioning $\mathbb{P}$?
>
> $\rightarrow$ We modified equation (4), which describes our distributional objective function. While our main objects are probability measures defined on the pre-softmax feature space, both certain/uncertain samples $(Y, X)$, and their corresponding probability measures $(\xi, \mu)$ are parameterized by neural networks with network parameters $\theta$. While the proposed functional $\mathcal{F}$ transforming $\mu$ to $\mathcal{F} \mu$ is non-parametric, only trainable parameters in equation (4) is neural networks with parameters $\theta$. Thus, the minimization task defined in equation (4) is taken over the network parameter space $\theta \in \Theta$.
>
> $------------------------------------------$
>
> $\textbf{ii-4)}$ In definition 1, what distance measure $d(\cdot, \cdot)$ are authors using? If the authors are not using more than two different notions of distances, why don't you just write it directly?
>
> $\rightarrow$  In Definition 1, inequality $\mathbb{E}_{\mu} d_E^2(x_0, x) < \infty$ is not a definition of distance that we use. This is a mathematical condition for probability measure that can be treated as a element of the Wasserstein space. In the paper, we only consider the Wasserstein space $\mathcal{P}_2(\mathbb{R}^d)$ equipped with the Wasserstein distance $\mathcal{W}_2(\cdot, \cdot)$. We think that overlapped definition (our mistake) in our first draft brings such a confusion.
>
> $------------------------------------------$
>
> $\textbf{ii-5)}$ Using same $d$ for distance and dimensions add unnecessary difficulty.
>
> $\rightarrow$ The overlapped definition is now clarified. In the revised version, $d$ stands for the dimensionality of pre-softmax space (i.e., number of classes) and $d_E$ stands for the Euclidean distance defined on this space.
>
> $------------------------------------------$
>
> $\textbf{ii-6)}$ In Eq. (4), again it is not clear over what the minimum is being taken over.
>
> $\rightarrow$ As we mentioned in ii-3), we clarified where/what the minimization is being taken over in the revised version.

---

> ### Author Response · Authors · 2020-11-12
> **Response to Reviewer#1, (2/3)**
>
> [On the motivation/rationale of the proposed method]
>
> $\textbf{iii)}$  I am curious what the general intuition behind the proposed approach is. Why is the "minimum transportation cost" map $\mathcal{F}$ a sensible method to transport the high-loss samples? Are we making any implicit assumptions on the nature of label-flipping operations, or perhaps the learning dynamics itself?
>
> $\rightarrow$ The motivation/rationale of our work comes from the $\textbf{empirical observations}$ in Section 4.1.  In this section, we have investigated and verified theoretically/empirically that there exists a strong correlation between the Wasserstein distance $\mathcal{W}_2( \mu , \nu)$ (i.e., distance between uncertain measure $\mu$ and most certain measure $\nu$) and network confidence for the test dataset. Specifically, we have shown that if the upper-bound of Wasserstein distance, $\varepsilon$ in equation (5) drastically increases, the test accuracy proportionally decreases according to $\varepsilon$ (as shown in Figure 2.).  We call it as 'diverging effect' in the paper. To fully approach/estimate the upper-bound $\varepsilon$ as shown in the left column of Figure 2., we used very tractable/interpretable representation in equations (11), (12), rather than directly estimating equation (5).
>
> As we (partially) understand the mathematical (Wasserstein geometric) behavior of uncertain/certain measure in Section 4.1, we tried to build a machinery (i.e., $\mathcal{F}$), which is fully non-parametric and computationally efficient by using gradient flow-based (SDE-based) stochastic models to control such ill-behaved sample dynamics. Specifically, the transport mapping $\mathcal{F}$ (our central object) is designed to control the upper-bound $\varepsilon$ in equation (5) to induce high confidence for testset. Figure 1 shows the conceptual illustration of transporting $\mu$ to $\mathcal{F} \mu$ to prevent diverging effect.
>
>
> Again, we clarify the advantages/central goal of this work.
>
> 1. We simulated SDE related to Ornstein-Ulenbeck process, which can be defined as a fully non-parametric way (as explained in equation (7) and Section C.3. in Appendix), thus our method does not require any $\textbf{additional trainable network parameter}$ due to the simplicity of SDE in (7).  Moreover, simulating equation (7) is computationally efficient while it is just for-loops in implementation code. The only computational burden is to calculate accumulated gradients of $X_{t = T}$, but we have shown that this burden is substantially low. Contrary to our method, many of previous methods including Co-teaching, JoCoR, DivideMix use dual networks to deal with noisy labels. This computational efficiency gives a practical advantage of our method when the model complexity is high.
>
> 2. In contrast to prior works which focus on building machinery rather than suggesting theoretical frameworks to understand the behavior of networks attacked by noisy labels, as we aforementioned in the introduction, the primal purpose of our work is to understand the geometric/statistical relation between certain and uncertain samples for corresponding probability measures $(\xi, \mu)$.  The most prominent results of our work is $\textbf{interpretability}$.  Proposition (2) makes us possible to empirically estimate the upper-bound of Wasserstein distance between certain and uncertain measure. For example, consider $t=0$. In this case, the uncertain measure is not transported as $\mathcal{F}_{t=0}\mu = \mu$. In addition, the Wasserstein distance is bounded by the combination of numerical constants $K_1(\nu)$ and $K_2(\mu)$ in equation (13) (There constants are defined in equations (25) and (40)). In general, the constant $K_1(\nu)$ is small compared to $K_2(\mu)$ (Please refer to equation (40) in Appendix. ), thus $\varepsilon \propto K_2(\nu) + K_2(\mu)$. The left column in Figure 2 shows that this value indicates whether the network is over-parameterized or not. If this value is properly controlled (we think that any methods based on small-loss criteria actually reduce this value), the network escapes from over-parameterization, as shown in right column in Figure 2. Proposition (3) clearly shows the maximal probability of uncertain measures being different to mean of certain measure in the label space is controlled with an inverse exponential ratio. As  mentioned by reviewer#4, we also would like to emphasize the novelty of Proposition (3), because, to the best of our knowledge, this type of statistical bound of robustness for noisy label has not been investigated before, although it renders the probabilistic relationship between $\mathcal{F}_t \mu$ and $\nu$.
>
> [[ The introduction section is modified to deliver more clarified motivation/contribution ]]

---

> ### Author Response · Authors · 2020-11-12
> **Response to Reviewer#1, (3/3)**
>
> $\textbf{i-1)}$ Missing baselines
>
> $\rightarrow$ We added important missing baselines into the revised paper in Section 2, which include [DivideMix], [INCV], [Arazo et al], and [Pleiss et al](NeurIPS 2020 paper). Briefly introduction is provided in Section 2.  Methodological differences are discussed in Section B in Appendix.
>
> $------------------------------------------$
>
> $\textbf{i-2)}$ Empirical comparison to DivideMix on CIFAR10/100, Clothing1M.
>
> $\rightarrow$ As recommended by reviewer#1, we added the empirical comparison to DivideMix on CIFAR10/100 using open-source code provided by authors (https://github.com/LiJunnan1992/DivideMix). Unfortunately, it was impossible to directly compare the proposed method with baseline methods in Table 1 because of the different experimental environments. Thus, for fair comparison to other baselines in Table 1, we modified the followings in DivideMix code to normalize experimental environments:
>
> 1. The base architecture is replaced by 9-layed CNN (https://github.com/bhanML/Co-teaching/blob/master/model.py), while all the baselines in Table 1 including our method use it.
>
> 2. In DivideMix, code for generating synthetic noise is slightly different from that of widely used open-source for generating symmetric/asymmetric noises (please refer to [cifar.py] in following link, https://github.com/bhanML/Co-teaching/blob/master/data/cifar.py). To normalize/unify noise labeled data, we replaced noise generation part by that of used in the above link.
>
> 3. All the other hyper-parameters are not modified.
>
> $------------------------------------------$
>
> [Performance comparison on CIFAR-10/100]
>
> CIFAR-10 with symmetric noise with 50% ratio:  DivideMix [ 81.13 +- .18 ] < WDN [ 82.89 +- .13]
>
> CIFAR-10 with asymmetric noise with 45% ratio:  DivideMix [ 68.93 +- .33 ] < WDN [ 76.12 +- .29]
>
> CIFAR-100 with symmetric noise with 50% ratio:  DivideMix [ 49.41 +- .25 ] >  WDN [ 48.45 +- .27 ]
>
> CIFAR-100 with asymmetric noise with 45% ratio:  DivideMix [ 34.24 +- .63 ] < WDN$_{cot}$ [  44.41 +- .37 ]
>
>
> As we modified some of their code including network architectures and noise generation schemes, we think that the performance of DivideMix above is not optimal because we did not fine-tuned the hyperparameters, which are important for boosting the performance. (warmup time, alpha, labmda_u, etc).
>
> $------------------------------------------$
>
> [Performance comparison on Clothing1M]
>
> DivideMix [74.76],
>
> WDN [74.75] (This value was originally reported in Table 3 before the rebuttal phase.)
>
> In Table 3, we added the result of DivideMix for test accuracy on the Clothing1M dataset. The performance gap is marginal (+- .01).
>
> $------------------------------------------$
>
> In the revised version, we provide additional experimental result on comparison of computational cost. The following table indicates the average training time for 5 epochs.
>
> DivideMix [34.41 +- .53 seconds]
>
> WDN [12.72 +- .08 seconds]
>
> Vanilla [ 11.43 +- .05 seconds]
>
> $------------------------------------------$
>
> Overall, as the experimental results clearly indicate, we believe that our method produces reasonable results compared to DivideMix while our method is fully non-parametric and computationally very efficient (in training time and additional memory). Additional results are now provided in the revised paper.
>
> $------------------------------------------$
>
> [On the minor suggestions]
> All the mentioned problems are now fixed in the revised paper.

---

> > ### Comment · AnonReviewer1 · 2020-11-19
> > **Quick note,**
> >
> > Dear authors,
> >
> > thank you for the reply, and the efforts made to respond to my concerns.
> >
> > To let you know: I just updated my score ($4 \to 5$) to reflect the resolved concerns on the experiment part. I still am not sure if I am persuaded by the "motivations," but this could be the fault on my side. Will check again with a fresher head.
> >
> > Best,
> > R1.

---

> > > ### Author Response · Authors · 2020-11-20
> > > **Response to Reviewer#1 : Additional comments.**
> > >
> > > We’re thankful to the reviewer for the comments and reconsideration on experiments section.
> > >
> > >
> > > We would like to add following comparison to help understand the rationale of proposed method.
> > > $------------------------------------------$
> > >
> > > [Methodological comparison between DivideMix and WDN]
> > >
> > > In DivideMix, the ground-truth label in training set for $\color{red} \text{unlabeled data}$ is guided by the clean probability produced by $\color{blue}\text{the other network}$. ($i.e.,  \hat{y} \leftarrow t y + (1-t) y_{pred}$), where $y_{pred}$ is average prediction of model.
> > >
> > > Our method can be considered to perform similar procedure, that is, the 'distribution' of $\color{red} \text{unlabeled data}$ (uncertain measure) is guided (transported) by proposed $\color{blue} \text{non-parametric transportation mapping}$ $\mathcal{F}$  ($i.e., \hat{\mu} \leftarrow \mathcal{F}_t \mu$).
> > >
> > > In this comparison, one can find following one-to-one corresponds :
> > >
> > > [Manipulated un-trusted object]
> > >
> > > $\hat{y} \leftrightarrow \hat{\mu}$,  (the guided unlabeled samples $\leftrightarrow$  the enhanced uncertain measure )
> > >
> > > [Semantic manipulation in different space]
> > >
> > > $ t y + (1-t) y_{pred}  \leftrightarrow \mathcal{F}_t \mu$,  (affine combination in label space $\leftrightarrow$ gradient flow in Wasserstein space)
> > >
> > > [Reference object as a trusted one for semantic manipulation]
> > >
> > > $y_{pred}  \leftrightarrow  \nu$.
> > >
> > >
> > > Thus, we think that the main motivation/methodology of WDN is similar to that of DivideMix. But, we'd like to emphasize the difference:
> > >
> > > As we know how the un-trusted samples behaves (Section 4.1), we can focus on building mathematical machinery to control such ill-behaved dynamics in vanilla network. In our work, the mathematical observations/models for ill-behaved samples comes first, and the statistical model ($i.e., \mathcal{F}$) is suggested as a outcome.
> > >
> > > In contrast, model-based approaches such as Co-teaching, DivideMix are highly dependent on the model-prediction by dual network because there are no prior models to explain the behavior of unlabeled samples.

---

### Official Review · AnonReviewer4 · 2020-10-28
**Recommendation to Accept**

**Rating:** 6
**Confidence:** 2

**Review:**

This paper propose a computationally efficient Wasserstein distributional normalization algorithm for accurate classification of noisy labels. An explicit upper bound for the Wasserstein-2 distance is derived and such a bound can be used as an estimator to determine if a network is over-parameterized. Empirical results on CIFAR-10/100 and Clothing1M suggest that the propose algorithm outperforms other SOTA approaches.

Overall, this paper is very well-written and easy to follow. The problem is well-motivated, the proofs looks solid, and the claims are well supported by experiments.

I am positive with respect to acceptance of this paper. First, compare to previous works, the proposed method (WDN) is fully non-parametric, sufficiently leverage the batches of training datasets, and do not assume any prior knowledge on class dependency. Hence WDN is flexible and potential has a smaller generalization error (compare to existing method based on sample selection). Second, I would like to emphasize the technical quality of the paper.  The concentration results in this paper are by themselves quite  elegance.

---

> ### Author Response · Authors · 2020-11-13
> **Response to Reviewer#4.**
>
> Thank you for taking the time to review the paper.
> As you noted, our paper is to give a understandable explanation of samples under small-loss criterion. We believe that extensive theoretical/empirical results in this work are key contributions to understand networks attacked by noisy labels. The future work should tackle more detailed mathematical issues in our paper.
>
> In order to address other reviewers comments, additional experimental results on CIFAR10 are added to the revised paper. To show the effectiveness of proposed method, we also added a additional 'computational cost' section to compare the required training time.

---

### Official Review · AnonReviewer2 · 2020-10-28
**Potentially a very good paper but writing could be drastically improved.**

**Rating:** 6
**Confidence:** 3

**Review:**

The paper is a contribution that aims at solving the label noise problem. In this setting, the labels are possibly corrupted, this yielding a potentially significant underperformance of a (neural network) classifier when minimizing the empirical risk. This problem is ubiquitous and important in real life scenarii. The paper builds on the idea of small loss criteria, which favors learning on certain samples in the beginning of the learning process, and gradually incorporate uncertain samples along iterations. The paper proposes a novel type of distributional normalization based on Wasserstein distance. It projects uncertain samples on a Wasserstein ball defined wrt. the certain samples. This process is done with a particle based stochastic dynamics, based on a Ornstein-Ulenbeck process. A theoretical Analysis is given, along with results on classical datasets in the symmetric noise setting, open noise and a real world dataset (clothing 1M), for which it achieves very good performances compared to state of the art competing methods.

While the overall idea is interesting, and the results described in the paper seem impressive and state-of-the-art on many of the tasks considered, I believe that the paper is badly written and very hard to follow. In the reviewer’s opinion, the major problems arise from:
 - a lack of introduction on the label noise problem, and a more comprehensive overview of the so called small loss criterion and its rationales;
 - despite an apparently rigorous mathematical framework, there are a lot of vague assertions (see minor remarks) and arbitrary choices (two examples: certain and uncertain distributions are looked upon in the pre-softmax logins, why ? Is the continuity equation of proposition 7 the only choice to model \mathcal{F} ? Some parts are really unclear (Why taking a detour ? what is the role of the Wasserstein moving average ? Is solving the SDE corresponding to the OU process the only solution to perform the normalization ? There are many algorithms in the literature that allows to estimate an OT mapping without solving this SDE)

Finally, whereas symmetric noise is considered in the experiment, I wonder how much the method is amenable to the more realistic setting of asymmetric noise, which is more likely to occur in the real life scenarii.

As a conclusion, I think there is a lot of interesting ideas and material in this paper, but the writing should be improved and clarified. I am willing to revise my note positively given that those aspects are taken into account in a revision of the paper.


Minor comments:
- why taking the pre-softmax activations to build \ mu and \epsilon ?
- some sentences are rather vague : ‘a functional that can manipulate \mu to induce geometric properties...’ ‘the pivot measure can be represented as a mini-batch containing the best combination of certain samples’ ‘the other iteration ‘ of which process ?
- Définition 1: ‘d’ is used for the distance symbol and the dimensions of the input signal
- Why Eq. 7 is a proposition ? It is rather a modeling choice
- Typo p8 CIAFAR-10 -> CIFAR-10

### After response to authors
I thank sincerely the authors for providing a detailed answer to my concerns. I changed my note to 6. I believe the paper is interesting and show strong empirical evidences that the method is worth considering. I am not giving a higher grade because in my opinion the writing of the paper could be signifcantly improved.

---

> ### Author Response · Authors · 2020-11-11
> **Response to Reviewer#2, (1/4)**
>
> $\textbf{i)}$ A lack of introduction on the label noise problem, and a more comprehensive overview of the so called small loss criterion and its rationales;
>
> $\rightarrow$ We have revised the introduction section to contain more comprehensive overviews of noisy label problems and the rationales of small-loss criterion based methods are provided in section (1), and (2).
>
> $------------------------------------------$
>
>
> $\textbf{ii-1)}$ Certain and uncertain distributions are looked upon in the pre-softmax logits, why?
>
> Why taking the pre-softmax activations to build $\mu$ and $\epsilon$?
>
> $\rightarrow$ In  most of previous baseline methods including Co-teaching, Co-teaching++, JoCoR, GCE, and SIGUA, which implement the small-loss criteria as their main methodologies, one of main issues to deal with in their methods is to suggest efficient sample selection strategies determining certain/uncertain samples, where these approaches are based on heuristic intuition rather than focusing on geometric constraints or distributional manipulation of semantic information to find finer probability concentration bounds of noisy labeled datasets. Thus, considering other spaces is unnecessary in their method. But, compared to these methods, our approach is based on imposing distributional/geometric constraint on uncertain samples using advanced statistical methods such as OT and Markov theory. To simplify the theoretical assumptions, and to explicitly define the SDE, we need well-studied spaces such like the Euclidean space $\mathbb{R}^d$, i.e, the pre-softmax feature space.
>
> [The technical difficulties for applying general OT theory to label space.]
>
> Let $X, Y$ be uncertain and certain samples, respectively, in the pre-softmax feature space. Assume that we consider the distributional constraint on the label-space (the space of $\sigma(X), \sigma(Y)$, where $\sigma$ denotes the soft-max function). This space is not proper to define the objective function such as (5). Because, all the samples in this label-space is of the form $\sigma(X) = [a_1, a_2, \cdots, a_n]$ such that $\sum^d_{i=1} a_i = 1$, thus the label-space is $d$-dimensional affine-simplex $U_d$, which is a subset of the Euclidean space $U_d \subset \mathbb{R}^d$. In this case, the definition of the Wasserstein space in equation (4) is unacceptable while $d_E$ is not a true metric on $U_d$. The Wasserstein space $\mathcal{P}_2(U_d)$ is merely investigated in the mathematical literature, which makes unable to use all the technical details, assumptions, and theories developed in the $\mathcal{P}_2(\mathbb{R}^d)$, which are theoretical ground of our work. But, we can look this problem slightly different point of view. For example, we can consider pre-softmax $\mathbb{R}^d, \mathcal{P}_2(\mathbb{R}^d)$ as our base space. In this case, all the technical issues/problems when we try to use OT tools in $\mathcal{P}_2(U_d)$ can be overcome/ignored. While softmax is non-parametric one-to-one function connecting pre-softmax feature space $\mathbb{R}^d$ to $U_d$, there exists an unique label in $U_d$ as a mapped point of the manipulated uncertain samples. Even though our objects are defined on pre-softmax space, the theoretical analysis in Proposition 3 contains a softmax function to evaluate the concentration inequality of the proposed transformation $\mathcal{F}$ affecting in the label-space $U_d$. Contrary to our method, [A] directly manipulates labels on $U_d$ in their methods by convex combination, which is simple and well-defined operation on $U_d$. This kind of approach is efficient for their method  while Bayesian model assumptions have to be considered rather than geometric manipulations on $U_d$.
>
> [[We added additional section in the Appendix commenting reasons for defining objects on pre-softmax space.]]
>
> $------------------------------------------$
>
> [A], Unsupervised Label Noise Modeling and Loss Correction,  Arazo et al, NeurIPS 2019.

---

> ### Author Response · Authors · 2020-11-11
> **Response to Reviewer#2, (2/4)**
>
> $\textbf{ii-2)}$ Why taking a detour?
>
> $\rightarrow$ We start by equation (4) as follows: Let us assume the extreme case such that the Wasserstein distance between most certain measure $\nu$ and manipulated uncertain measure $\mathcal{F}_t \mu$ is zero for some $t > 0$, i.e., $\mathcal{W_2}(\mathcal{F}_t \mu, \nu) = 0$. In this case, it is trivial that $\mathcal{F}_t \mu = \nu$, and equation (4) turns into the following form:
>
> $\min_{\sim} \mathcal{J}[\nu] + \mathcal{J}[\xi] = \min_{\sim} \mathcal{J}[\xi_{k^{\star}}] + \mathcal{J}[\xi_k]$ for some $k^{\star}$.
>
> The equality holds because we define that $\nu$ is just the one of the possible $\xi_{k^{\star}}$ for past certain measures for $k^{\star} \in [1, \cdots, k-1]$, where $k$ indicates current learning iterations for training time. If we train our network with above equation, only 'certain' samples are considered during training time, and thus the network falls into over-parameterization quickly. Now, let us assume the opposite extreme case such that $\mathcal{W_2}(\mathcal{F}_t \mu, \nu) \rightarrow \infty$ for some $t > 0$. In this case, the random variable related to $X_t \sim \mathcal{F}_t \mu$ can be of an arbitrary form, which is unacceptable in classification tasks.
>
> Thus, balancing the upper-bound $\varepsilon$ is a crucial designing factor for our method. To deal with such a problem without additional constraints or assumptions to avoid increase model complexity, we simply set a detour in equation (5) by triangle inequality of the Wasserstein distance. Thus, the original distance $\mathcal{W}_2(\nu, \mathcal{F}\mu)$ is divided into two additional terms in equation (5). The first term $\mathcal{W}_2(\nu, \mathcal{N})$ term makes us not to consider the collapsing case $\varepsilon \rightarrow 0$, if the most certain measure $\nu$ does not change over time, this term is constant and non-zero if and only if $\nu$ is not a Gaussian measure (which is hardly the case in a high dimensional feature space). Now, we only need to consider the second term to control the upper bound $\varepsilon$. The main advantage of considering the second term as of form $\mathcal{W}_2(\mathcal{N}, \mathcal{F}\mu)$ is that we can control it by using a simple iterative equation (7), which is fully non-parametric and computationally efficient to control the upper-bound $\varepsilon$, as shown in equations (5) and (7).

---

> ### Author Response · Authors · 2020-11-11
> **Response to Reviewer#2, (3/4)**
>
> $\textbf{ii-3)}$ Is the continuity equation of proposition 7 the only choice to model $\mathcal{F}$?
> ii-4) Is solving the SDE corresponding to the OU process the only solution to perform the normalization?
>
> $\rightarrow$ If one of equations (7) or (6) is defined first, another equation is uniquely defined automatically. In other words, there exists a one-to-one correspondence between SDE and PDE of the form (7), and (6). Selecting a proper type of (6) is equivalent to selecting the corresponding SDE of the form (7). Thus, for simplicity, we only focus on SDE-based interpretation to answer both  ii-3) and ii-4).
>
> There are reasons for selecting SDE of OU process (7) as our main stochastic dynamics and their corresponding PDE defined in (6).
>
> [On the practical point of view for advantage of OU process: computational time, non-parametric representations]
>
> 1. We tried to build simple, computationally efficient, and non-parametric method to estimate/minimize the second term of equation (5). The OU process is one of the most simplest SDE with fixed drift and diffusion terms over times $t$, which induces the non-parametric representations of Euler-Maruyama scheme, as shown in (7), which also ensures the convergence of scheme (7) (Please refer to C.3 in Appendix. detailed information is commented). If the some other time-inhomogeneous process is selected as a baseline stochastic dynamics such that the drift and diffusion terms varies over times, one may need additional parameters to fully represent time-varying characteristics of SDE, which can increase the model complexity.
>
> 2. Due to the simplicity of the non-parametric form of the OU process in (7), GPU-based computation for simulation is very cheap, while the total amount of computational time of simulating iterative equation (7) is substantially low as it is just for-loops for $T$-times in the implementation code. Please refer to our comment of 'computational efficiency'
>
> [On the theoretical point of view for selecting OU process: uniqueness, accessibility to empirical evidence]
>
> 3. If the detour measure is set to the fixed Gaussian measure (Note that our detour Gaussian measure is only dependent on $\nu$, while the path of stochastic dynamics of (7) is dependent on $\mu$), the 'unique' Feller process (generalization of OU process with  Lipschitz continuous drift, diffusion terms) is the OU process.
>
> 4. One of main contributions of our work is to show a $\textbf{strong empirical evidence}$ such that the properly controlled upper-bound $\varepsilon$ in equation (5) actually helps the network from escaping over-parameterization. In Section 4.1, both equations (11), and (12) are proposed to empirically estimate the upper bound $\varepsilon$ to be controlled, which is only efficiently estimated when the process is the OU-type (The left-hand side of ineqaulity (12) is called Mehler's formula in mathematics literature, which can be efficiently estimated thanks to the expectation form). If we select other types of Feller process as stochastic dynamics inducing $\mathcal{F}$, we are not able to estimate the observation such like the upper-bound landscape, as shown in the left column of Figure 2, while there's no canonical formula for explicitly estimating the upper-bound in the general Feller process. Please refer to Proposition 4 in Appendix for detailed mathematical considerations.
>
> Despite the aforementioned considerations, we can still select any generalized Feller process and its corresponding SDE, but the most desirable properties such as interpretability and computational efficiency will be canceled out.
>
> [[We briefly added comments of reason/motivation for selecting OU process in the revised paper.]]
>
> $------------------------------------------$
>
> $\textbf{ii-5)}$ Why Eq. 6 is a proposition? It is rather a modeling choice?
>
> $\rightarrow$ As  mentioned by reviewer#2, equation (6) and its corresponding stationary distribution (i.e., $d\mathcal{N}_{\nu} = dq_t dx)$  is a modeling choice. But the main argument of Proposition (1) is not showing equation (6) is satisfied but the Wasserstein distance $\textbf{exponentially reduced}$ from an initial length as $t \rightarrow \infty$. (i.e., $\mathcal{W}( \mathcal{N}_{\nu}, \mathcal{F}_t \mu) \propto e^{-t}$) if we select PDE in (6). This means that the second term in equation (5) can be controlled with an inverse exponential ratio. This also gives theoretical advantages by putting the detour measure by Gaussian (related to OU process).

---

> ### Author Response · Authors · 2020-11-11
> **Response to Reviewer#2, (4/4)**
>
> $\textbf{ii-6)}$  What is the role of the Wasserstein moving average?
>
> $\rightarrow$ Wasserstein moving average is an additional trick for boosting performance of the proposed method. While most certain measures $\nu$ are updated up to a current learning iteration $k$, in the experiments of easy tasks (e.g., CIFAR10 with 20% symmetric noise), we have observed that $\nu$ is not updated for a while after 10 epochs. This may leads undesirable results when $\nu$ is updated so that the Wasserstein distance of current certain measure $\xi_k$ and $\nu$ is drastically changed. The proposed method reduces the impact of sudden change of $\nu$ by applying moving average of Gaussian parameters when updating $\nu$.
>
> $------------------------------------------$
>
> $\textbf{ii-7)}$ There are many algorithms in the literature that allows to estimate an OT mapping without solving this SDE.
>
> There are many types of empirical methods to estimate OT costs (especially Wasserstein distance) in the machine learning literature. The first type of method implements the dual formulation of 1-Wasserstein distance, which is widely used adversarial network models. The second type of method implements the primal formulation of Wasserstein distance by directly searching the discretized probability coupling using sinkhorn projection. However, these two methods require additional machinery (primal form) or learnable parameters (dual form) to properly estimate the OT costs. In contrast, gradient flow-based (or SDE-based) methods such as [B] and our method are very fast to implement and non-parametric. In training with such complex data and network (e.g., training ResNet50 with Clothing1M dataset), additional heavy computation is not desirable. Thus, we think that the last type of method is desirable for our use.
>
> $------------------------------------------$
>
> $\textbf{iii)}$ I wonder how much the method is amenable to the more realistic setting of asymmetric noise, which is more likely to occur in the real life scenarii.
>
> $\rightarrow$ Actually, we conducted the experiments with asymmetric noise. The noise type of 'pairflip' is completely identical to 'asymmetric' noise. We used same code for generating both symmetric and asymmetric synthetic label noise in open-source code of Co-teaching network. We will change the notation 'pairflip' to 'asymmetric'.
>
> $------------------------------------------$
>
> [Minor comments]
>
> $\textbf{ⅳ-1)}$ Why taking the pre-softmax activations to build $\mu$ and $\xi$?
>
> $\rightarrow$ We clarified the theoretical reasons in ii-1), and methodological difference between our method and [A].
>
> $\textbf{ⅳ-2)}$ 'A functional that can manipulate \mu to induce geometric properties...'
>
> $\rightarrow$ We clarified the geometric property of manipulated (or normalized) uncertain measure $\mathcal{F}_t \mu$ in ii-5). We will modify the sentence in the revised paper.
>
> $\textbf{ⅳ-3)}$  ‘The pivot measure $\nu$ can be represented as a mini-batch containing the best combination of certain samples...’
>
> $\rightarrow$ We will modify the sentence in the revised paper.
>
> $\textbf{ⅳ-4)}$ Definition 1: ‘d’ is used for the distance symbol and the dimensions of the input signal...
>
> $\rightarrow$ Now, $d$ stands for the dimensionality of pre-softmax feature space, and $d_E$ stands for the Euclidean distance defined on this space.
>
> $\textbf{ⅳ-5)}$ Typo p8 CIAFAR-10 -> CIFAR-10
>
> $\rightarrow$ We will modify the sentence in the revised paper.
>
> $------------------------------------------$
>
> [A], Unsupervised Label Noise Modeling and Loss Correction, Arazo et al, NeurIPS 2019.
>
> [B] Sliced-Wasserstein Flows, Liutkus et al., ICML 2019.

---

### Official Review · AnonReviewer3 · 2020-10-30
**Obscure motivation of proposed objective**

**Rating:** 4
**Confidence:** 4

**Review:**

Summary

The paper introduces a novel objective function by imposing geometric constraints on the logits of uncertain samples. The authors' approach is to map the distribution logits of uncertain samples onto the 2-Wasserstein ball centered on the measure of certain samples. To overcome the dilemma of selecting the ball radius, the authors propose a surrogate objective, namely Wasserstein Normalization. An SDE grad flow is proposed for solving the Wasserstein normalization. The paper also keeps the Gaussian parameters as moving average during training in light of batch normalization. The paper both theoretically and empirically validate their method.

Contributions

i) Proposal of a novel Wassersterin Normalization objective for uncertain samples.

ii) Theoretically giving a verifiable upper bound of the constrain term.

iii) Extensive experiments on synthetic and real-world datasets.

Issues:

i) The motivation of normalizing the uncertain measures into the 2-Wasserstein ball of certain measure is unclear to me. And it seems irrelevant to the practical problem. Yes, I agree that we should prevent the over-parameterized network to overfit the uncertain samples since the majority of them are noisy samples (network tends to fit noisy samples slowly). Based on this observation, the idea of discarding/relabeled uncertain samples is widely adopted in this field. The paper's approach belongs to this category. But the motivation of Wasserstein Normalization for uncertain samples is weak. I cannot directly see its benefits. Imagining the network has high confidence for certain samples (which is generally true in practice), then the mean of gaussian $m$ is uniform categorical distribution.  The paper's approach is just injecting noise onto the network output of uncertain samples.

ii) Does the simulation of the SDE makes the training much slower? Since we need the calculate the grad flow for every single sample during every iteration of the inner loop.

iii) Could the authors better explain the dilemma of selecting the ball radius $\epsilon$ in Sec 3? I cannot see why the benefits decrease when $\epsilon\approx 0$.

---

> ### Author Response · Authors · 2020-11-11
> **Response to Reviewer#3, (1/2)**
>
> $\textbf{i-1)}$ Motivation of our method. [[ According to following answers, we modified motivation section in revised version ]]
>
> $\rightarrow$ As a result of our observation and extensive ablation study developed in Section 4.1, we theoretically and empirically verified that the partitioned probability measure $\mu$ of uncertain samples starts to diverge from the certain probability measure $\xi$ in the Wasserstein space (as shown in left column of Figure 2) when we train with noisy labeled data.
>
> The crucial part of this section is that we found the strong correlation between diverging effect and classification performance with noisy labels. In particular, if the Wasserstein distance becomes too large such that the upper-bound of (5) is diverged(upper bound of distance between $\nu$ and $\xi$, i.e., $\mathcal{W}_2(\nu, \xi) \rightarrow \infty$), the classification performance rapidly decreased  (as shown in right column of Figure 2).
>
> In light of this observation, one needs to "control" the uncertain measure $\mu$ to alleviate such a diverging effect for accurate classification results. Thus, our main motivation is to develop,  (a) theoretically grounded and (b) computationally effective control method to deal with noisy labeled data (as shown in Figure 1-b).
>
> (a) Most previous works implementing small-loss criteria are the lack of theoretical analysis. To understand the behavior of certain/uncertain samples beyond heuristic intuitions from algorithms, we used the mathematical tools from optimal transport (OT).  To the best of our knowledge,  the analytic representations of data certainty like inequalities in equation (9) and (11) in Proposition [2, 3] are not rigorously explored before. In addition, this geometric phenomenon/relation between certain and uncertain measures are not observed before in our community, even though it is a potential key to understand over-parameterization trained with noisy labeled data.
>
> (b) Our method is fully 'non-parametric' and fast to implement. Please refer to our response to 'computational efficiency'.
>
> $------------------------------------------$
>
> $\textbf{i-2)}$ Imagining the network has high confidence for certain samples (which is generally true in practice)...
>
> $\rightarrow$ The definition of certain samples in many existing works including our method varied over the number $\rho N$ of certain samples, where $\rho$ indicates the portion of certain samples over 'total training samples' $N$. No matter how we select the ratio $\rho$, the training accuracy (confidence of training samples) can be high as mentioned by reviewer#3, if the network is arbitrary large. However, the main problem of classification tasks with noisy labels is that there's no explicit/canonical relation between training accuracy and test accuracy according to varying noise types. In most cases, 'high' training accuracy (confidence) in vanilla networks (with cross-entropy loss) inevitably leads 'low' test accuracy, if we train them with noisy labeled data (The red dotted line in Figure 1-(b) indicates this phenomenon).
>
> Our work is to understand such a veiled relation between training accuracy and test accuracy with mathematical tools using equation (7) or (12).
>
> $------------------------------------------$
>
> $\textbf{i-3)}$ The mean of Gaussian $\mathbf{m}$  is uniform categorical distribution...
>
> $\rightarrow$ Please note that the first and second moments (mean and covariance, i.e., $\mathbf{m}, \Sigma$) are defined on the best certain measure $\nu = \arg \min \mathcal{J}[\xi_{k^{\star}}]$. If the network is so certain about training samples (high confidence on training samples), we also agree that the distribution of $\nu$ will be very similar to the Gaussian mixture with $d$-number of modes, where the mean of each nodes can be understood as a categorical random variable. However, as aforementioned in i-2), we cannot evaluate or assume the statistical structure of the distribution with "noisy labeled test dataset". It can have a very long-tail property compared to the distribution of training samples or it may have a form which cannot be represented as a simple Gaussian mixture model. In this case, any method based on such assumptions may fail to predict behavior of noisy samples. For example, [A] also assumes the mixture model but it fails to learn with the Clothing1M dataset, which has non-uniform noise types.
>
> In this circumstance, our method is relatively free in assumptions on data distributions. As  mentioned by reviewer#4, we do not assume any 'prior knowledge' about the data. The only required pre-assumption of the proposed method is to select the stochastic dynamics of gradient flow,  where in our case the OU-process and its corresponding Fokker-Planck equation are used, making our method computationally efficient and non-parametric.
>
> $------------------------------------------$
>
> [A], Unsupervised Label Noise Modeling and Loss Correction, Arazo et al, NeurIPS 2019.

---

> ### Author Response · Authors · 2020-11-11
> **Response to Reviewer#3, (2/2)**
>
> $\textbf{i-4)}$ The paper's approach is just injecting noise onto the network output of uncertain samples.
>
> $\rightarrow$ As aforementioned by reviewer#3, our method injects noise to uncertain samples, which seems to be arbitrary and not novel at first glance. But, more careful attention should be paid because the way of injecting noise in the equation (7) is to simulate stochastic differential equations (SDEs). Specifically, the Gaussian noise is iteratively injected to simulate SDEs of Ornstein-Ulenbeck type in equation (7), which ensures that the probability measure of noise-injected uncertain samples  $\mu_t \sim [X_{t}^n]$ are gradually closer to certain measure $\xi$, as $t$ grows to prevent the diverging effect (which is exactly the our motivation). (Here, $X_{t}^i, X_{t}^j$ is assumed to be i.i.d. for any $i \neq j$.)
> Please refers to Section C.3 in Appendix for detailed analysis.
>
> In this circumstance, equation (4) indicates that our classification network is trained with both certain and 'manipulated' uncertain samples (for $t = T$ iterations) to prevent over-parameterization in a fully non-parametric way.
>
> $------------------------------------------$
>
> $\textbf{ii)}$ Does the simulation of the SDE makes the training much slower? Since we need the calculate the grad flow for every single sample during every iteration of the inner loop.
>
> $\rightarrow$ In our implementation and GPU-based computation, the computational complexity of simulating iterative equation in (7) is substantially low, because it is nothing but for-loops $T$-times in the implementation code, and the iterative summation is calculated in gpu-based batch-wise manner. The only burden is to calculate the gradient of manipulated uncertain samples summed up to $T$-times. However, we have observed that the total amount of time of calculating gradients is very low even although $T$ is large. i.e., $T=64$. Please refer to Section 5.3 in the revised paper.
>
> Please note that our baseline methods (Co-teaching, Co-teaching++, and JoCoR) use 'dual' networks, thus require (x2) computational complexity (network parameters, training time, etc.) to train.
> In contrast, our method only uses a single base vanilla network with a low extra computational burden.
> In addition, $\textbf{our method does not use any additional trainable network parameters}$. This computational efficiency is one of our practical advantages. Please refer to added the 'computational cost' section in the revised paper.
>
> $------------------------------------------$
>
> $\textbf{iii)}$ Could the authors better explain the dilemma of selecting the ball radius $\varepsilon$ in Section 3?  I cannot see why the benefits decrease when $\varepsilon \approx 0$.
>
> $\rightarrow$ We start by equation (4) as follows:
> Assume the much extreme case such that the Wasserstein distance between most certain measure $\nu$ and manipulated uncertain measure $\mathcal{F}_t \mu$ is zero for some $t > 0$, i.e., $\mathcal{W_2}(\mathcal{F}_t \mu, \nu) = 0$. In this case, it is trivial that $\mathcal{F}_t \mu = \nu$. (Technically, it is true because there always exists a unique geodesic in the Wasserstein space connecting $\nu$ and $\mathcal{F}_t \mu$ equipped with the Riemannian distance with $L_2$ type and $\mathcal{F}_t \mu$ is always lying in the 2-Wasserstein space by the conservativeness of continuity equation (6). Please refer to [C. Villani. Optimal Transport: Old and New] ). Thus, the equation (4) can be changed into the following form:
>
> $\min_{\sim} \mathcal{J}[\nu] + \mathcal{J}[\xi] = \min_{\sim} \mathcal{J}[\xi_{k^{\star}}] + \mathcal{J}[\xi_k]$ for some $k^{\star}$.
>
> The equality holds because we define that $\nu$ is just the one of the possible $\xi_{k^{\star}}$ for past certain measures for $k^{\star} \in [1, \cdots, k-1]$, where $k$ indicates current learning iterations for training time. If we train our network with the above equation, only 'certain' samples are considered during training time, and thus the network falls into over-parameterization quickly in light of comments on i-2) and i-3). Now, assume the opposite extreme case such that $\mathcal{W_2}(\mathcal{F}_t \mu, \nu) \rightarrow \infty$ for some $t > 0$ (This can be also attained because the 2-Wasserstein space is not compact in general). In this case, the transformed uncertain samples related to $X_t \sim \mathcal{F}_t \mu$ can be arbitrary form (while 'distance' is very large) which is unacceptable in a classification task.
>
> Thus, balancing $\varepsilon$ is crucial part of the proposed method. To deal with the aforementioned problem without additional constraints or assumptions, we need to set a detour in equation (5), which is nothing but triangle inequality of the distance. It ensures that $\varepsilon$ prevents overly enlarged distances or zero distances, while the first term in (5) is mathematically ensured not to be zero. This is central rationales of setting detour.

---

### Official Review · AnonReviewer5 · 2020-12-03
**A good idea but the paper needs to be extremely more polished**

**Rating:** 4
**Confidence:** 3

**Review:**


This paper aims to deal with the learning of noisy/corrupted labels based on the small loss criterion. If I understood well the idea is to consider a new loss function on the Wasserstein space to learn the certain and uncertain data distributions. This loss function is based on kind of penalty term ensuring that the uncertain labels lies in a Wasserstein ball for which the radius is automatically tuned to get the best possible result. This construction crucially relies on the use of the Wasserstein gradient flow associated with Gaussian distributions. To conclude, a series of experiments show that the new methodology proposed in the paper leads to state-of-art results.

I think that the idea is nice and was very impressed by the numerical results reported by the authors. However, it took me a while to just understand the problem and the setting considered by the authors! My second main criticism concerns the theoretical results presented in this paper. While I think that it is important that methods are justified by rigorous results, the one presented in this paper are just not understandable by most people. Two of the main underlying issues in my opinion are that the notion and objects which are used in the paper are not introduced very much rigor (or even not at all) and the result lack of clarity. To be honest, I did not catch half of the sentences of the paper.

I advise the authors to make a in-depth revision of their paper, introducing more carefully their method so it can be understand by a broader audience. One solution in my opinion, is to reduce the theoretical notion and results  to their strict minimum. I think that a lot of them are unnecessary for the introduction of the proposed methodology.

---

### Author Response · Authors · 2020-11-11
**Comments on computational efficiency**

We added 'computational cost' to Section 5.3 in the revised paper. While the non-parametric methods such as GCE and WDN require  additional computational time less than $12$%, other methods that require additional networks spent more time compared to non-parametric methods. In the following table, time was measured based on public source code provided by authors. We used a single GTX 2080ti GPU for all experiments.

Method | Vanilla | GCE | WDN | Co-teaching | JoCoR | DivideMix |

-----------------------------------------------------------------------------------------------
Time          |   11.43   |  11.53 |  12.72 |   15.88   |   17.88  |   34.41  |

---

### Author Response · Authors · 2020-11-16
**Comments on additional/modified results.**

In order to address reviewers comments, we modified followings in revised version.

1. The Section [1, 2] are significantly modified as follows:

    1-1) We clarified motivations/rationale of our method, to emphasize that motivation is based on the empirical observation in Section  4.1. This shows that controlling 'diverging effect' of certain/uncertain measures is key factor to deal with noisy labels.

    1-2)  We added important missing baselines into the revised paper, which include [DivideMix], [INCV], [Arazo et al], and [Pleiss et al],  Methodological differences are discussed in Section B in Appendix.

    1-3) We added comprehensive overview of noisy-labels, and small-loss criterion.


2. Motivations of Wasserstein Normalization

    2-1) The reason for setting detour as Gaussian, using OU-process is much clarified.  In the section 3.1, theoretical/empirical advantages are emphasized in boldface fonts.

    2-2) The reason for defining probability measure on pre-softmax space is provided. (Section C, Appendix)

3. Additional experiments

    3-1) We added 'computational cost' section in the revised paper to show the efficiency of our method.

    3-2) Additional experimental results on CIFAR-10/100 are added.

4. The title is changed  to give detailed information about the task under consideration:
    $\rightarrow$'Wasserstein distributional normalization: Nonparametric stochastic modeling for handling noisy labels'.

5. All the minor issues are now solved.

---

### Decision · Program_Chairs · 2021-01-07
**Final Decision**

**Decision:**

Reject

**Comment:**

This paper proposes a potentially very interesting and original approach to handle label noise.  The numerical experiments suggest that the method works very well. But the paper  itself has been deemed very hard and demanding to read and understand for a general machine learning crowd and even by experts in the fields of optimal transport and  Markov theory.

Note that due to the low confidence in several review an additional emergency review by an expert was asked and it confirmed the global opinion from  other reviewers that the paper is interesting but needs a major rewriting before acceptance in a ML conference. The AC strongly suggest that the authors work on a more pedagogical introduction and explanation of the method before resubmitting.